# Cooperation between a hierarchical set of recruitment sites targets the X chromosome for dosage compensation

Sarah Elizabeth Albritton, Anna-Lena Kranz, Lara Heermans Winterkorn, Lena Annika Street, Sevinc Ercan*

Department of Biology, Center for Genomics and Systems Biology, New York University, New York, United States

**Abstract** In many organisms, it remains unclear how X chromosomes are specified for dosage compensation, since DNA sequence motifs shown to be important for dosage compensation complex (DCC) recruitment are themselves not X-specific. Here, we addressed this problem in *C. elegans*. We found that the DCC recruiter, SDC-2, is required to maintain open chromatin at a small number of primary DCC recruitment sites, whose sequence and genomic context are X-specific. Along the X, primary recruitment sites are interspersed with secondary sites, whose function is X-dependent. A secondary site can ectopically recruit the DCC when additional recruitment sites are inserted either in tandem or at a distance (>30 kb). Deletion of a recruitment site on the X results in reduced DCC binding across several megabases surrounded by topologically associating domain (TAD) boundaries. Our work elucidates that hierarchy and long-distance cooperativity between gene-regulatory elements target a single chromosome for regulation.

## Introduction

Eukaryotic genomes are large, encompassing thousands of genes and spanning across millions of base pairs. In order to fit within the limiting confines of the nucleus, genomic DNA is compacted in a hierarchical manner. Linear DNA wraps around histones to form nucleosomes. A string of nucleosomes makes up the chromatin fibers, which further organize into variably sized topologically associating domains (TADs) (*Dekker and Heard, 2015*; *Matharu and Ahanger, 2015*). Finally, each chromosome occupies an individual territory within the nucleus (*Dekker and Mirny, 2016*; *Denker and de Laat, 2016*). Importantly, this genome organization is a dynamic process with implications for transcriptional regulation. For example, nucleosome positioning regulates transcription factor (TF) binding at promoters and enhancers, which in turn interact via long-range chromatin looping. At the domain-scale, the coordinated regulation of many genes across hundreds of kilobases of DNA has been observed in several organisms (*Cohen et al., 2000*; *Scott-Boyer and Deschepper, 2013*; *Spellman and Rubin, 2002*). For instance, the coordinated regulation of gene clusters, such as the *Hox* genes, can span hundreds of kilobases and requires the specific recruitment of regulatory proteins (*Bauer et al., 2016*). How gene regulatory complexes are specifically targeted to large chromosomal domains remains poorly understood. Research on the X-specific targeting of dosage compensation machinery in various animals helps to elucidate the molecular mechanisms that specifically target chromosomal domains for transcriptional co-regulation.

X chromosome dosage compensation refers to mechanisms that equalize X chromosome gene expression between males (XY) and females (XX) of diploid species wherein sex is determined by differences in X chromosome copy number (*Adler et al., 1997*; *Albritton et al., 2014*; *Chen and Zhang, 2015*; *Deng et al., 2011*; *Lin et al., 2012*, *Lin et al., 2011*; *Veitia et al., 2015*;

*For correspondence: se71@nyu.edu

**Competing interests:** The authors declare that no competing interests exist.

**eLife digest** The DNA inside living cells is organized in structures called chromosomes. In many animals, females have two X chromosomes, whereas males have only one. To ensure that females do not end up with a double dose of the proteins encoded by the genes on the X chromosome, animals use a process called dosage compensation to correct this imbalance. The mechanisms underlying this process vary between species, but they typically involve a regulatory complex that binds to the X chromosomes of one sex to modify gene expression.

*Caenorhabditis elegans*, for example, is a species of nematode worm in which individuals with two X chromosomes are hermaphrodites and those with one X chromosome are males. In *C. elegans*, a regulatory complex, called the dosage compensation complex, attaches to both X chromosomes of a hermaphrodite, and reduces the expression of the genes on each by half to match the level seen in the males. Previous research has shown that short DNA sequences, known as motifs, recruit the dosage compensation complex to the X chromosomes. However, these sequences are also found on the other chromosomes and, until now, it was not known why the complex was only recruited to the X chromosomes.

Albritton et al. now show the X chromosomes have a 'hierarchical' recruitment system. A few sites on the X chromosomes contain clusters of a specific DNA motif, which initiate the process and attract the dosage compensation complex more strongly than other sites. These 'strong' recruitment sites are placed across the length of the X chromosomes and cooperate with several 'weaker' ones located in between. This way, multiple recruitment sites can cooperate over a long distance, while non-sex chromosomes, which have only one or two stronger recruitment sites, do not have thisadvantage.

Hierarchy and cooperativity may be general features of gene expression, in which proteins are targeted to chromosomes without the need for having specific motifs at every recruitment site. The way DNA sequences are distributed across the genome may give us clues about their role. Thus, knowing how genomes are structured will help us identify disrupted areas in diseases such as cancer.

*Wheeler et al., 2016*). Here, we focus on the transcriptional regulatory mechanisms that act to restore X expression balance between the sexes. In wild type mammals, X expression balance is achieved via X inactivation, wherein one of the two female X chromosomes is transcriptionally silenced during development (*Heard and Disteche, 2006*). In *Drosophila melanogaster,* the male-specific lethal (MSL) complex binds to the single X chromosome in males where it upregulates transcription two-fold (*Conrad and Akhtar, 2012*). In *Caenorhabditis elegans*, both hermaphrodite X chromosomes are targeted by the dosage compensation complex (DCC), resulting in a two-fold downregulation of expression from each (*Ercan and Lieb, 2009*; *Ercan, 2015*). Although the mechanisms are diverse, in each case, a protein complex is specifically targeted to the X chromosome in only the sex where it regulates transcription (*Ercan, 2015*).

The working model for X-specific binding of the different dosage compensation complexes involves a two-step strategy: recruitment and spreading (*Ercan, 2015*). While the mechanism of spreading has been observed to be sequence-independent (*Ercan et al., 2009*; *Kelley et al., 1999*; *Sun and Birchler, 2009*), in both *C. elegans* and *D. melanogaster*, recruitment is accomplished in part by short DNA sequence motifs (*Alekseyenko et al., 2012*, *Alekseyenko et al., 2008*; *Ercan et al., 2007*; *Jans et al., 2009*). Although important for DCC and MSL recruitment, the DNA sequence motifs are not sufficient to explain X-specificity. In both species, motif sequence is found across the genome, and is only slightly enriched on the X chromosome (*Alekseyenko et al., 2012*, *Alekseyenko et al., 2008*; *Ercan et al., 2007*; *Jans et al., 2009*). This epitomizes an important open question in general mechanisms of domain-scale gene regulation: how are gene regulatory complexes specifically targeted to large chromosomal domains when their corresponding recruitment motifs lack domain-specificity. In this study, we combined systematic ChIP-seq analysis with recruitment mutants and targeted genome editing to determine how the *C. elegans* DCC is specifically recruited to the X chromosomes. Our work offers an answer to the specificity question, indicating

that hierarchy and long-distance cooperation between a set of motif-containing recruitment sites restricts binding of the DCC to the X chromosomes.

At the core of the DCC is a condensin complex (hereafter condensin DC) (*Csankovszki et al., 2009a*, *Csankovszki et al., 2009b*) (*Figure 1A*). Condensins are evolutionarily conserved protein complexes, most often cited for their role in chromosome condensation and segregation during cell division (reviewed in [*Hirano, 2016*]). Recent work also suggests key roles for condensins in gene regulation during interphase (*Cobbe et al., 2006*; *Kranz et al., 2013*; *Longworth et al., 2012*; *Rawlings et al., 2011*; *Dej et al., 2004*; *Lupo et al., 2001*). Condensins are composed of a dimerizing pair of structural maintenance of chromosomes proteins (SMC-2 and SMC-4) that interact with three chromosome-associated polypeptides (CAPs). *C. elegans* condensin DC shares four out of five subunits (MIX-1 [*Lieb et al., 1998*], DPY-26 [*Plenefisch et al., 1989*], DPY-28 [*Plenefisch et al., 1989*], and CAPG-1 [*Csankovszki et al., 2009b*]) with the canonical condensin I, distinguished only by the SMC-4 variant, DPY-27 (*Csankovszki et al., 2009a*, *Csankovszki et al., 2009b*). Condensin DC interacts with at least five non-condensin proteins, SDC-1,2,3, DPY-30, and DPY-21, which together form the genetically defined DCC (*Meyer, 2005*). With the exception of SDC-1 and DPY-21, all DCC subunits are essential (*Plenefisch et al., 1989*; *Villeneuve and Meyer, 1990*). DPY-30, in addition to its role in dosage compensation, is a subunit of the highly conserved MLL/COMPASS complex, which methylates histone H3 at lysine 4 (H3K4) (*Pferdehirt et al., 2011*; *Li and Kelly, 2011*; *Shilatifard, 2008*; *Hsu and Meyer, 1994*).

Fluorescence microscopy using DCC-specific antibodies indicated that the DCC binds to both hermaphrodite X chromosomes (*Chuang et al., 1996*; *Dawes et al., 1999*). Subsequent high-resolution ChIP-chip and ChIP-seq experiments revealed a pattern of DCC binding that supports the recruitment and spreading hypothesis (*Ercan et al., 2009*, *Ercan et al., 2007*; *Jans et al., 2009*). The recruitment sites show high levels of DCC binding, while sites of spreading show comparatively weaker DCC binding and frequently overlap with promoters and enhancers (*Ercan et al., 2009*, *Ercan et al., 2007*; *Kranz et al., 2013*). DCC spreading is independent of X chromosome sequence as the complex is able to spread into autosomal sequence fused to the end of the X (*Ercan et al., 2009*). Interestingly, condensin DC spreads more effectively than the recruiter proteins SDC-2 and SDC-3 (*Ercan et al., 2009*), highlighting the distinction between recruitment and spreading. Recruitment of the DCC to the X chromosomes is dependent on SDC-2, SDC-3, and DPY-30: SDC-3 binding requires both SDC-2 and DPY-30 (*Davis and Meyer, 1997*); DPY-30 binding requires both SDC-2 and SDC-3 (*Pferdehirt et al., 2011*). Only SDC-2 can localize to the X chromosomes in the absence of other complex members (*Dawes et al., 1999*).

Identification of DNA sequences capable of autonomously recruiting the DCC has previously been conducted using extrachromosomal arrays (*Csankovszki et al., 2004*; *McDonel et al., 2006*). Briefly, the DNA fragment to be tested is injected into the worm where it forms a multi-copy array. Fluorescence microscopy is then utilized to assess DCC recruitment to the extrachromosomal sequence. However, the array-based assay is limiting in both function (in that it is not feasible to test every possible X chromosome sequence) and in specificity (in that assaying recruitment to artificial, multi-copy arrays can result in false positives). Here, to overcome these limitations, we used a genomics approach to identify and categorize DCC recruitment sites in the context of the native X chromosome. To better understand how a limited number of recruitment sites function to specifically target the DCC to the X, we investigated the role of DNA sequence and chromatin accessibility on recruitment site activity, and performed targeted genome editing to test recruitment site function.

Here, we found that DCC recruitment to the X chromosomes is initiated by a small number of strong recruitment sites that contain clusters of a 12 bp DNA sequence motif. Strong recruitment sites display significant overlap with annotated high occupancy transcription factor target (HOT) sites and are marked by DNA-encoded nucleosome occupancy, two features that distinguish motif clusters on the X from those on the autosomes. Strong recruitment sites are distributed along the length of the X chromosome, interspersed by weaker sites whose recruitment function is X-chromosome dependent. We addressed the X-dependent activity of recruitment sites by ectopically inserting recruitment sequence into both the autosomes and the X chromosome. While the ectopic insertion of a recruitment site in single copy failed to fully recruit the DCC to an autosome, both increased copy number and the insertion of additional recruitment sequences at a distance significantly increased DCC recruitment to an ectopic site, suggesting that cooperation between multiple recruitment sites is necessary for robust DCC binding on the X. Furthermore, deletion of a recruitment site

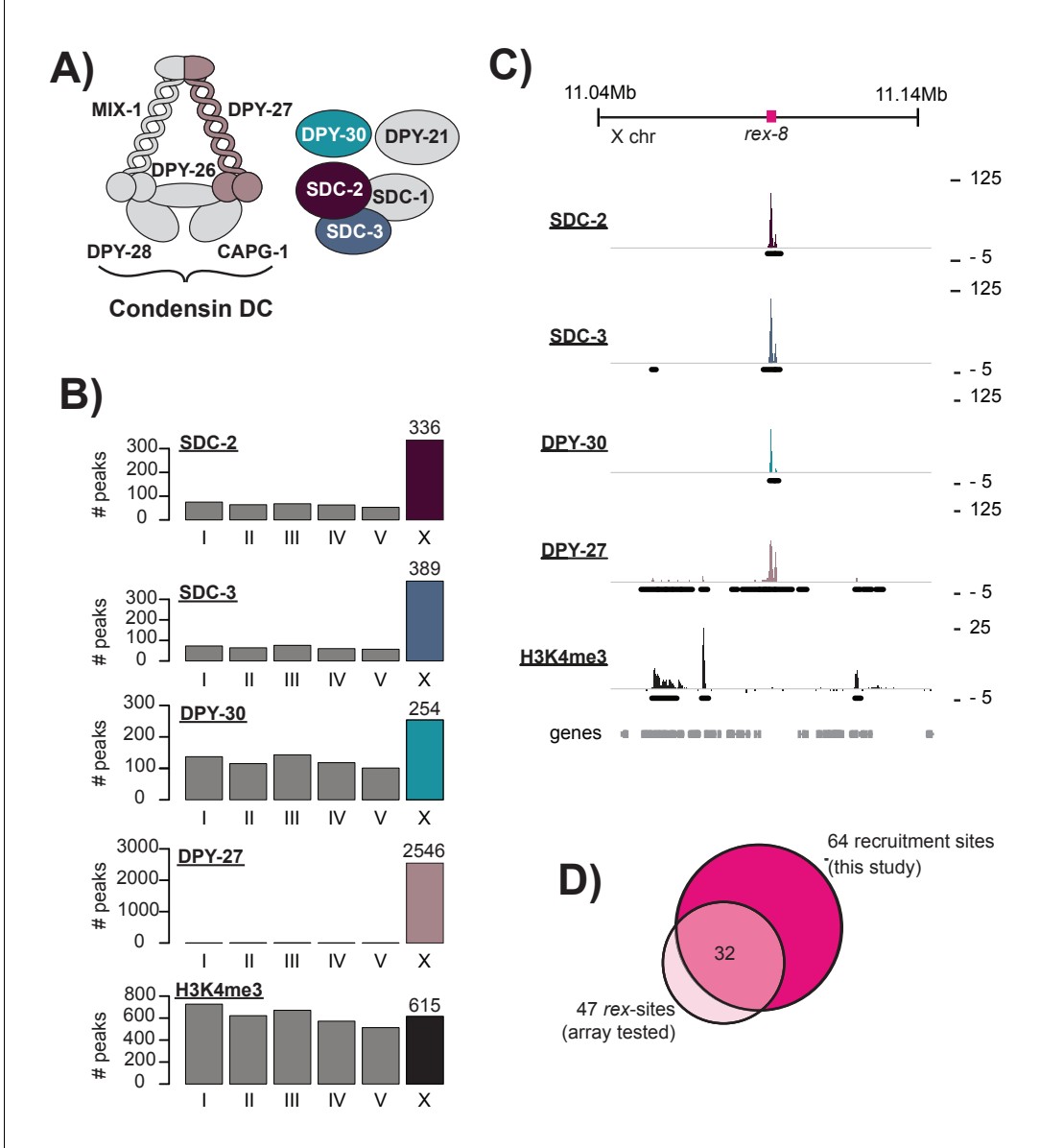

**Figure 1.** DCC recruitment sites are defined using high resolution ChIP-seq analysis. (**A**) The *C. elegans* dosage compensation complex (DCC) is composed of a modified condensin complex (condensin DC) which is distinguished from condensin I by the SMC-4 variant, DPY-27. The non-condensin subunits SDC-2, SDC-3, and DPY-30 are required for DCC recruitment to the X chromosomes. (**B**) Peak distribution across each of the six chromosomes. (**C**) Representative ChIP-seq enrichment for an 80 kb window that includes recruitment element on the X, *rex-8*. Recruitment sites on the X chromosome were defined as a 400 bp window centered on the SDC-2 ChIP-seq summits that overlap with SDC-3, DPY-30, and DPY-27 peaks and that do not show significant H3K4me3 enrichment. (**D**) Recruitment sites identified in this study (n = 64) largely overlap with sequences previously shown to recruit DCC to multi-copy extrachromosomal array (n = 47) (*Jans et al., 2009*; *Pferdehirt et al., 2011*; *McDonel et al., 2006*).

The following figure supplements are available for figure 1:

**Figure supplement 1.** ChIP-seq data suggests that some previously defined rex-sites fail to recruit the DCC in the context of the X chromosome.

**Figure supplement 2.** Peaks excluded by H3K4me3 overlap do not resemble strong recruitment sites.

resulted in moderate but significant reduction of DCC binding centered on the deleted site and surrounded by strong TAD boundaries. This result suggests that recruitment by multiple sites and subsequent spreading of the DCC in cis within distinct chromosomal domains establishes the full DCC binding pattern. Overall, our work demonstrates that cooperation between a hierarchical set of recruitment sites specifically targets the *C. elegans* dosage compensation complex to the X chromosomes.

## Results

### Determining DCC recruitment sites using ChIP-seq data

Previously, extrachromosomal array based assays were used to demonstrate the ability of a DNA fragment to autonomously recruit the DCC, independent of the X chromosome. This technique was used to identify 47 recruitment elements on the X, named *rex-1* through rex-47 (*Jans et al., 2009*; *Pferdehirt et al., 2011*; *McDonel et al., 2006*). However, extrachromosomal arrays carry multiple copies of the sequence being tested leaving these assays susceptible to false positives. Many sequences shown to recruit the DCC to a multi-copy extrachromosomal array show little to no DCC binding in ChIP-seq experiments (*Figure 1—figure supplement 1A*). Here, to identify sequences that recruit the DCC in the context of the native X chromosome, we used *C. elegans* mixed developmental stage embryos (~100 cell stage and higher obtained by bleaching adults and using similar growth conditions between strains) to perform ChIP-seq analysis of SDC-2, SDC-3, and DPY-30, the proteins required for DCC recruitment (*Pferdehirt et al., 2011*; *Dawes et al., 1999*; *Davis and Meyer, 1997*). Consistent with previous work (*Ercan et al., 2009*), we found that SDC-2 binding is enriched on the X chromosome: 336 out of 659 (51%) binding sites are on the X. Similarly, 54% (389/719) of SDC-3 and 29% (254/868) of DPY-30 binding peaks are on the X chromosome. The condensin DC portion of the complex, as assayed by DPY-27 ChIP-seq (*Kranz et al., 2013*), is largely restricted to the X chromosome: over 99% (2546/2564) of DPY-27 binding peaks are on the X. Peak distribution (*Figure 1B*) and a representative window of ChIP-seq enrichment for SDC-2, SDC-3, DPY-30, DPY-27, and H3K4me3 (a marker of COMPASS activity [*Shilatifard, 2008*, *Shilatifard, 2012*; *Xiao et al., 2011*]) (*Figure 1C*) are shown. As a side note, among the 336 autosomal SDC-2 binding peaks, 206 (61.3%) overlap with both SDC-3 and DPY-30 but fail to recruit condensin DC, suggesting a DCC-independent role for the recruiter proteins at these sites.

Previous work has demonstrated that high levels of DCC ChIP enrichment are predictive of recruitment activity (*Jans et al., 2009*). Thus, in order to define a list of DCC recruitment sites, we began by identifying the 100 strongest SDC-2 ChIP-seq binding peaks in the genome. Of these, 92 overlap with both SDC-3 and DPY-30 binding peaks. DPY-30, as a member of MLL/COMPASS, is associated with the histone modification H3K4me3 (*Shilatifard, 2008*, *Shilatifard, 2012*; *Xiao et al., 2011*). To remove sites of COMPASS activity, we eliminated peaks that overlap with H3K4me3 enrichment. Lastly, we looked for the presence of condensin DC binding by requiring overlap with a DPY-27 binding peak on the X chromosome. Our criteria for defining recruitment sites was strict; 9 DCC binding loci on the X chromosome were eliminated due to high enrichment of H3K4me3 (*Figure 1—figure supplement 2*). Our finalized list of 64 recruitments sites contains 32 out of 47 (68%) *rex*-sites that were shown previously to recruit the DCC to extrachromosomal arrays (*Figure 1D*, *Supplementary file 1D*). The remaining 15 previously identified *rex* sites either lack binding of one or more DCC subunits (14/15) and/or overlap with H3K4me3 enrichment (6/15), indicating COMPASS activity. For clarity, when referring to loci identified by our ChIP-seq enrichment method we will use the term 'recruitment site.' When referring to a specific sequence that has been previously shown to recruit the DCC to array we will use the term '*rex*-site.' ChIP-seq enrichment at recruitment sites, both endogenous and ectopic, will be referred to as 'DCC recruitment.' ChIP-seq enrichment at all other genomic loci will be referred to as 'DCC binding.'

### SDC-2 recognizes and binds a set of strong recruitment sites in the absence of other DCC members

Closer examination revealed that ChIP enrichment scores at the recruitment sites vary greatly; some sites appear to recruit the DCC much more robustly than others (*Figure 1—figure supplement 1B*). Recruitment sites were ranked by their SDC-2, SDC-3, DPY-30, and DPY-27 ChIP enrichment scores

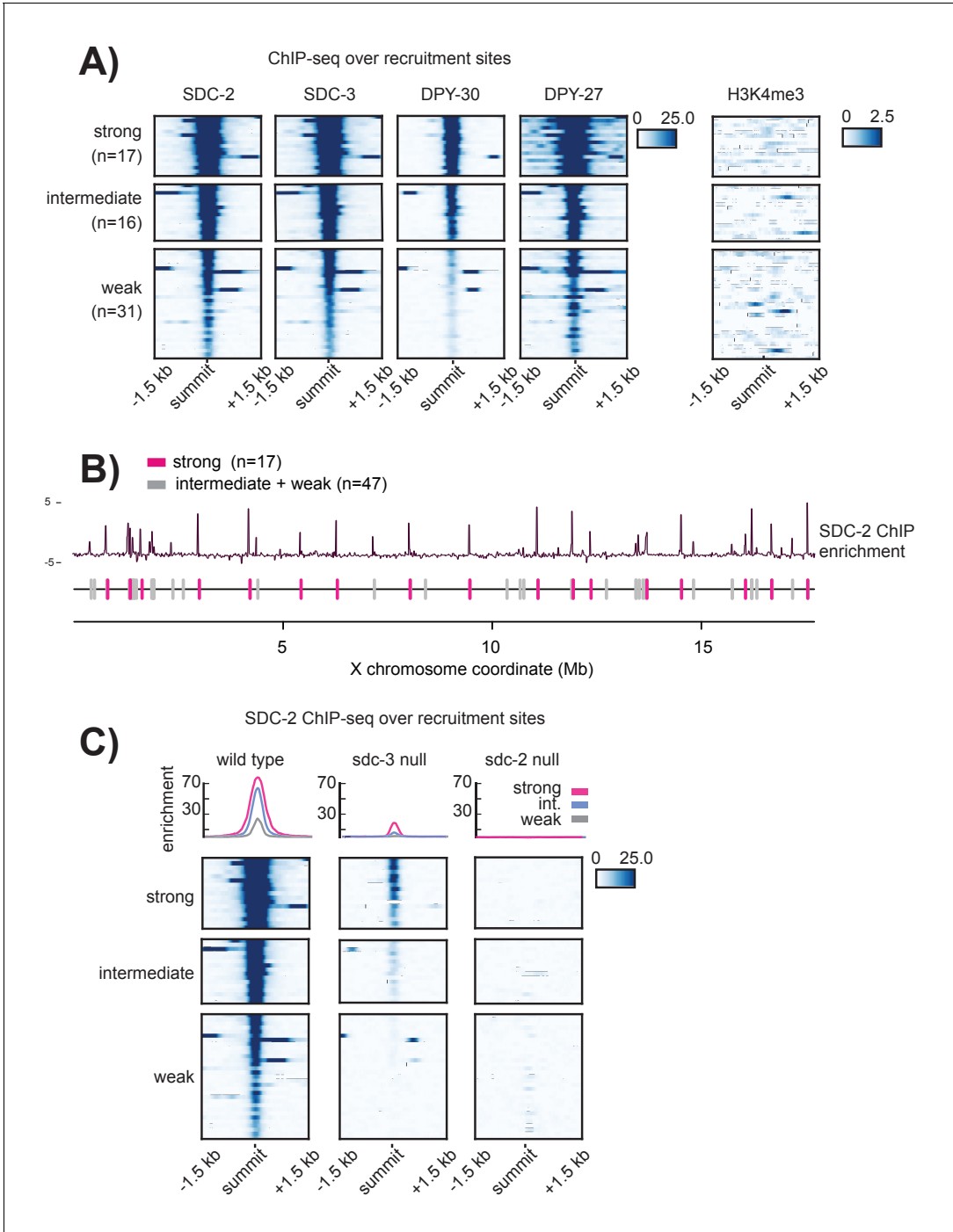

**Figure 2.** Recruitment sites on the X chromosome are hierarchical (**A**) Heatmaps showing ChIP-seq enrichment for the DCC recruiter proteins, SDC-2, SDC-3, and DPY-30, the condensin DC subunit DPY-27, and the histone modification H3K4me3. Heatmaps are plotted across a 3 kb window centered on the 64 recruitment sites. Based on the DCC ChIP enrichment scores, the recruitment sites are categorized into three strength classes (strong, intermediate, weak) using k-means clustering (n = 3). (**B**) The strong DCC recruitment sites (pink, n = 17) are scattered across the X chromosome. Intermediate (n = 16) and weak (n = 31) recruitment sites (gray) are distributed in between. The median distance between recruitment sites is ~90 kb. SDC-2 enrichment across the length of the X chromosome was plotted using IGV. (**C**) In an *sdc-3* null mutant, SDC-2 binds the strongest recruitment sites, indicating hierarchy of recruitment. As in (**A**), heatmaps indicating SDC-2 ChIP-seq enrichment are plotted across a 3 kb window centered on the 64 recruitment sites. Data from wild-type, *sdc-3* null, and *sdc-2* null are all plotted on the same scale. Density plots above each heatmap indicate the average ChIP enrichment score for each recruitment class plotted across the 3 kb window. Strong sites are plotted in pink, intermediate in blue, and weak in grey.

*Figure 2 continued on next page*

*Figure 2 continued*

The following figure supplement is available for figure 2:

**Figure supplement 1.** Specific reduction in SDC-2 enrichment at weak recruitment sites in an sdc-3 null mutant.

and separated into three strength classes using k-means clustering (*Figure 2A*). We identified 17 strong, 16 intermediate, and 31 weak recruitment sites. Strong recruitment sites are spaced roughly 1 Mb apart along the length of the X chromosome and are interspersed by both weak and intermediate strength sites (*Figure 2B*). This observed pattern of recruitment site localization and strength suggested that X-specific recruitment of the DCC is hierarchical, initialized by DCC binding at a subset of strong recruitment sites.

Among the DCC subunits, only SDC-2 can localize to the X chromosome in the absence of other complex members and only SDC-2 is required for proper localization of all other dosage compensation proteins (*Dawes et al., 1999*). Under a model of recruitment site strength hierarchy, we predicted that the strongest recruitment sites might be acting as initial DCC entry sites. If correct, then SDC-2 ChIP-seq performed in an *sdc-3* null mutant strain (in which only SDC-2 can localize to the X) should reveal binding only at the strongest recruitment sites. Indeed, in *sdc-3 (y126)* null mutant embryos, SDC-2 binds to 16 out of 17 strong, 11 out of 16 intermediate, and only 2 out of 31 weak recruitment sites (*Figure 2C*). The single strong site that fails to recruit SDC-2 is in a region that is deleted in the *sdc-3* null mutant.

To test the possibility of an equivalent reduction of SDC-2 binding across all sites on the X chromosome in the *sdc-3* mutant, we plotted SDC-2 enrichment across the X chromosome in wild-type animals versus the *sdc-3* mutant (*Figure 2—figure supplement 1A*). We observe a striking difference between changes in SDC-2 binding at the strong versus the weaker recruitment sites. Across the length of the X chromosome, the background SDC-2 enrichment is reduced (median $\log_2$ ratio of $-1.280$, *Figure 2—figure supplement 1B*). Weak recruitment sites exhibit a markedly lower $\log_2$ ratio (median $-3.042$). In contrast, strong recruitment sites have a median $\log_2$ ratio of $-1.057$. Retainment of SDC-2 enrichment specifically at the strong recruitment sites in animals lacking *sdc-3* is consistent with hierarchical binding of SDC-2 at recruitment sites.

Additionally, the pattern of SDC-2 ChIP-seq enrichment in the *sdc-3 null* suggests that SDC-2 binding is restricted to the initial recognition sites. Compared to wild-type, we see only a narrow window of SDC-2 binding in the *sdc-3 null* strain, indicating that SDC-2 fails to spread in the absence of other DCC members (*Figure 2C*). As a control, ChIP-seq analysis in *sdc-2 (y74)* null mutant embryos revealed a complete lack of SDC-2 binding at all 64 recruitment sites (*Figure 2C*). In sum, these observations suggest that our categorization of recruitment sites into stronger and weaker classes using the pattern of DCC ChIP enrichment reflects the hierarchy of recruitment. Strong recruitment sites are the initial entry points for DCC binding, which allows for DCC binding at weaker recruitment sites.

## SDC-2 is required to maintain open chromatin at strong recruitment sites, which display intrinsic DNA-encoded nucleosome occupancy

SDC-2 is a novel 344.4kD protein unique to *Caenorhabditis* (*Nusbaum and Meyer, 1989*) and the only hermaphrodite-specific member of the DCC. All other complex subunits are maternally loaded into the embryo (*Dawes et al., 1999*; *Stoeckius et al., 2014*) and remain diffuse in the nucleus prior to the onset of *sdc-2* expression at around the 40 cell stage (*Dawes et al., 1999*). While we cannot exclude the possibility that, prior to *sdc-2* expression, some condensin DC is loosely associated with all chromosomes, previous work indicates that SDC-2 is required for X-specific enrichment of condensin DC binding (*Dawes et al., 1999*). Despite its importance in recruiting the DCC to the X chromosomes, it remains unclear how SDC-2 is able to specifically recognize and bind to the X. Toward understanding the mechanism underlying X-recognition by SDC-2, we examined histone occupancy at recruitment sites, reasoning that initial recruitment to stronger sites might be the result of those sites being inherently more open prior to the onset of *sdc-2* expression. We performed histone H3 ChIP-seq in wild-type and *sdc-2* null mutant embryos, using the *sdc-2* null strain as a proxy to measure nucleosome occupancy before the onset of *sdc-2* expression. While wild-type H3 data indicated

a tendency for open chromatin at all recruitment sites (*Figure 3A*, left), we observed a marked increase in H3 enrichment, specifically at the strong recruitment sites, in the *sdc-2* null mutant (*Figure 3A*, right). Interestingly, a nucleosome occupancy model trained on in vitro sequence preferences (*Ercan et al., 2011*) predicts high nucleosome occupancy at DCC recruitment sites (*Figure 3B*). These results suggest that SDC-2 is required for open chromatin at the recruitment sites which themselves encode for high nucleosome occupancy.

## The 12 bp DNA sequence motif and its nucleosomal context have a role in DCC recruitment

Previous work using either array-characterized *rex*-sites (*Jans et al., 2009*; *McDonel et al., 2006*) or DCC ChIP-chip data (*Ercan et al., 2009*, *Ercan et al., 2007*) independently identified a short DNA sequence motif enriched at DCC recruitment sites. Extrachromosomal arrays bearing wild-type motif sequence were shown to recruit the DCC while arrays with motif mutations failed to recruit (*McDonel et al., 2006*), indicating that motif sequence is important for DCC binding. However, the motif is only slightly enriched (~3 fold) on the X (*Ercan et al., 2007*; *Jans et al., 2009*), and copies of the motif on autosomes do not recruit the DCC. To investigate the role of the motif in X-specific recruitment of the DCC, we first redefined the 12 bp motif using higher resolution ChIP-seq data

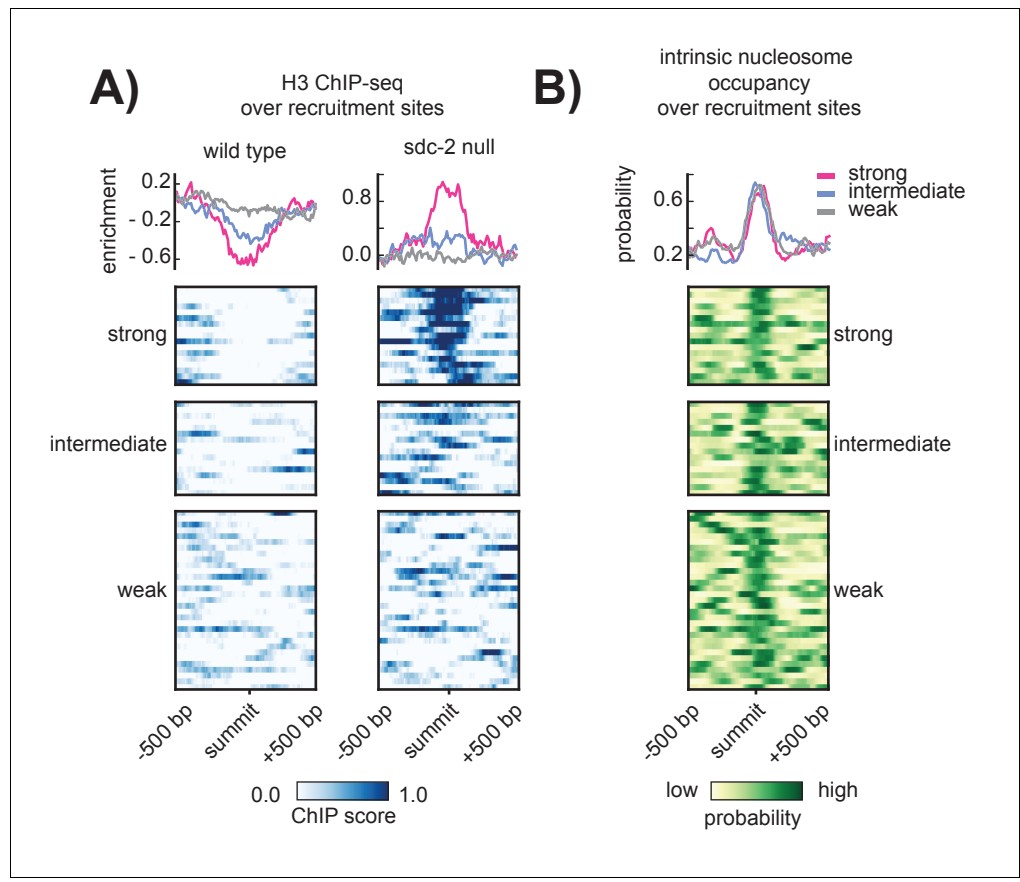

**Figure 3.** SDC-2 is required for open chromatin at strong DCC recruitment sites that encode for high intrinsic nucleosome occupancy. (A) Heatmaps showing H3 ChIP-seq enrichment across a 1 kb window centered on the 64 recruitment sites. Wild-type data (left) indicates open chromatin at all recruitment sites. Data from an *sdc-2* null strain (right) indicates increase in H3 enrichment, especially at the strong recruitment sites. Density plots above each heatmap indicate the average ChIP enrichment score for each recruitment class. (B) Heatmap indicates the probability of nucleosome occupancy across a 1 kb window centered on the recruitment sites. Probability ranges from 0.0 (yellow) to 1.0 (green). DNA-encoded nucleosome occupancy signal is highest at the center of the recruitment sites.

(*Figure 4A*). Similar to the previously identified sequences, the 12 bp motif contains an 8 bp GCGCAGGG core with variable specificities for the surrounding nucleotides. We scanned the whole genome for motif occurrences using the Transcription Factor Affinity Prediction (TRAP) tool (*Thomas-Chollier et al., 2011*), generating a genome-wide list of motif match locations and associated affinity scores (*Supplementary file 1E*).

We found that X-enrichment of the 12 bp recruitment motif is sensitive to affinity score cut-off. Motifs with a score of less than seven are randomly distributed throughout the genome while those with a score of seven or greater are approximately three fold enriched on the X. The X chromosome contains 43% (47/110) of motifs with a score of 8 or better, 64% (20/31) with a score of 9 or better, and 91% (10/11) of perfect match motifs (ATCGCGCAGGGA) (*Figure 4B*). Based on this enrichment

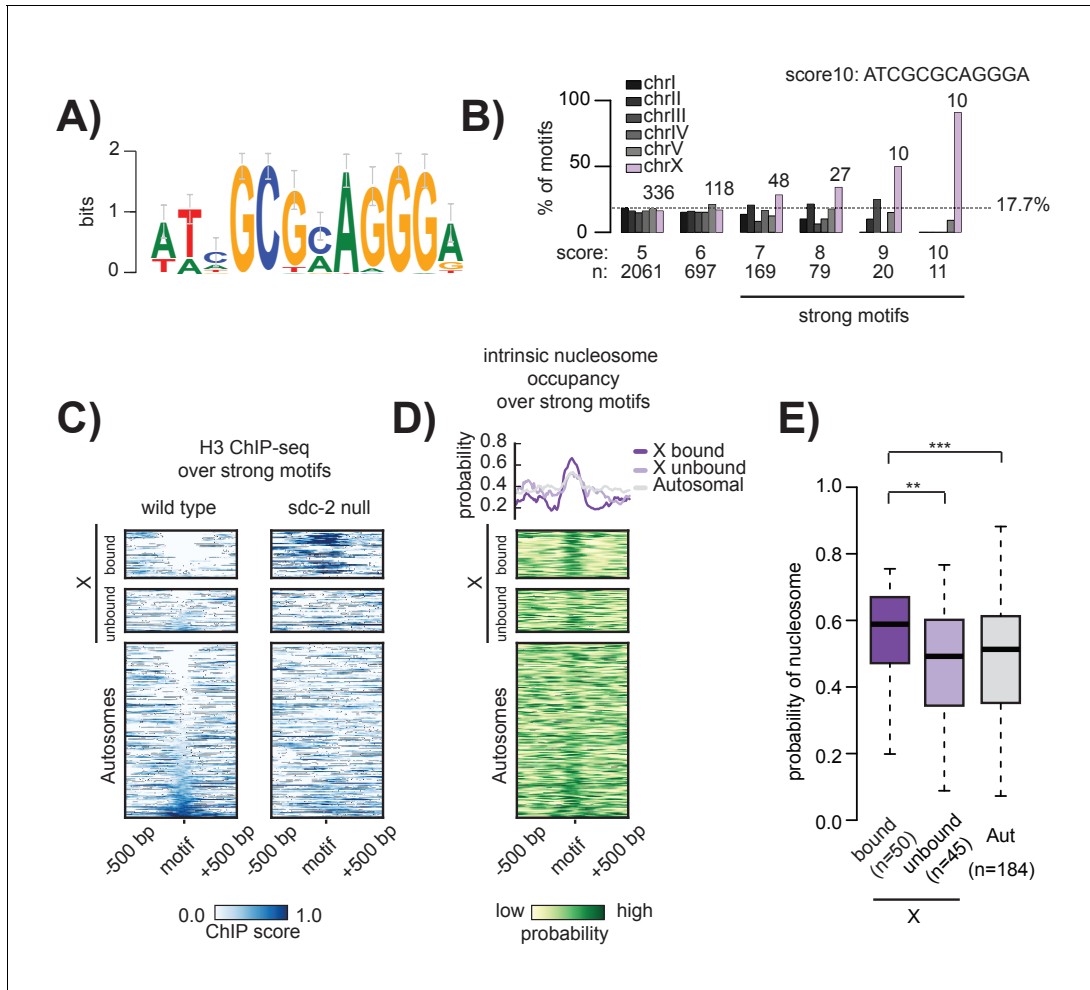

**Figure 4.** The 12 bp recruitment motif is enriched on the X. Bound motifs are characterized by DNA-encoded nucleosome occupancy. (**A**) The 12 bp recruitment motif identified using high-resolution SDC-2 ChIP-seq data. (**B**) Motif distribution is plotted for chromosomes I through V (shades of gray) and chromosome X (purple). Motif enrichment is dependent on score-cutoff. Weaker motifs (score <7) are randomly distributed across the chromosomes. Strong motifs (score ≥7) are enriched on the X chromosome. Perfect score motifs (score 10, sequence ATCGCGCAGGGA) show almost complete X-specificity: 10/11 (91%) are on the X. (**C**) Heatmaps showing H3 ChIP-seq enrichment from wild-type (left) and *sdc-2* null (right) embryos, plotted across a 1 kb window centered on all strong motifs of score ≥7. Motifs are sorted by H3 enrichment in wild-type, and divided into three categories: Bound by DCC on the X (n = 50), unbound on the X (n = 45), and unbound on the autosomes (n = 184). Here, we define 'bound' as the overlap of both SDC-3 and DPY-27 binding. There are no bound motifs on the autosomes. (**D**) Heatmap indicates the probability of nucleosome occupancy across a 1 kb window centered on the motifs. As in (**C**), motifs are sorted by H3 enrichment in wild-type. (**E**) Boxplot indicates intrinsic nucleosome occupancy for a 150 bp window centered on each motif. Bound motifs (dark purple, median 0.616) have significantly higher DNA-encoded nucleosome occupancy compared to unbound motifs on the X (light purple, median 0.4865) and motifs on the autosomes (grey, median 0.503). Significance calculated using one-tailed students t-test (** p-value<0.01, *** p-value<0.001).

pattern, we define strong motifs to be those with a score of 7 or better. Strong motifs are enriched on, but not specific to, the X chromosome. There are 184 strong motifs distributed across the five autosomes, none of which are bound by the DCC. And while all motifs with a perfect score of 10 recruit the DCC in the context of the X chromosome (*i.e.* all of them overlap with a recruitment site), the single perfect autosomal motif fails to recruit the DCC (*Figure 5—figure supplement 1*). Additionally, we find that, of the 95 strong motifs on the X chromosome, only 50 (52%) are bound by the DCC. This result parallels the general observation that many TFs bind only a subset of their potential binding motifs (*D'haeseleer, 2006*).

Because chromatin accessibility often functions to regulate TF binding, we reasoned that unbound motifs on both the X and the autosomes might be precluded from DCC binding due to the presence of nucleosomes. To test this, we plotted histone H3 ChIP-seq enrichment across 1 kb windows centered at all of the strong motifs in the genome (n = 279) (*Figure 4C*). We found that bound motifs are, indeed, depleted of histones. However, several motifs on both the X chromosome and the autosomes exhibit similar histone depletion, yet are not bound by the DCC. Therefore, motif accessibility is insufficient to predict DCC binding.

Because we observe an increase in H3 enrichment at strong recruitment sites in the absence of SDC-2 (*Figure 3A*), we reasoned that SDC-2 might be similarly required for open chromatin at strong motifs. In the absence of SDC-2, H3 occupancy increases specifically at motif sequences that are bound by the DCC in wild-type animals (*Figure 4C*). Increased H3 occupancy reflects the nucleosome occupancy predicted by the *in vitro* model (*Figure 4D*). Interestingly, DCC bound motifs have, on average, higher predicted nucleosome occupancy than unbound motifs on either the X or the autosomes (*Figure 4E*). Combined with our previous observation that recruitment sites are characterized by high intrinsic nucleosome occupancy (*Figure 3A*), our data suggests a role for nucleosome positioning in DCC binding and recruitment.

## Motif clustering and overlap with HOT sites confer X-specificity to the strong DCC recruitment sites

While enriched on the X chromosome, the 12 bp motif is also found on all five of the autosomes. However, unlike on the autosomes where motifs are more randomly dispersed, motifs on the X tend to cluster together in linear space (*Ercan et al., 2007*). Here, we refined the analysis of motif clustering by defining a motif cluster to be a strong motif (score ≥7) flanked within 200 bp by at least one other motif with a score of 5 or better. Using this definition, we identified 17 homotypic motif clusters on the X chromosome, all of which are bound by the DCC and all of which are contained within a recruitment site (*Figure 5A*). Significantly, of the 10 perfect motif matches on the X chromosome, nine are contained within a motif cluster and are centered at a strong recruitment site. Two of the strongest recruitment sites, *rex-14* and *rex-32*, each contain two perfect motifs (score = 10). Overall, motif clustering is a characteristic of strong recruitment sites. Among the 17 strong sites, six contain a cluster of two motifs and five contain a cluster of three motifs. These observations suggest that motif clustering on the X is important for DCC recruitment. However, clustering alone cannot fully distinguish the X from the autosomes. Using our definition, we find five motif clusters on the autosomes, none of which recruit the DCC (*Figure 5—figure supplement 1*). All five autosomal clusters have only one strong motif and one weaker motif. In addition, each autosome contains at most, two motif clusters (*Figure 5A*).

Alone, neither motif sequence nor motif clustering nor nucleosome occupancy is able to explain X-specificity of DCC recruitment. Subsequently, in order to identify additional factors that distinguish DCC recruitment sites on the X, we looked for significant overlap with various genomic annotations (*Figure 5B*). References for annotations can be found in *Supplementary file 1F*, and a more exhaustive overlap plot can be found in *Figure 5—figure supplement 2*. Previous work has demonstrated significant enrichment of condensin I binding at tRNA genes in yeast, chicken and *C. elegans* (*Kranz et al., 2013*; *D'Ambrosio et al., 2008*; *Haeusler et al., 2008*; *Kim et al., 2013*). Of the 638 annotated *C. elegans* tRNA genes, 286 (44.8%) are on the X chromosome. Of these, 178 (62.2%) overlap with a condensin DC binding site on the X chromosome, as assayed by DPY-27 ChIP-seq. Interestingly, 16 out of 64 DCC recruitment sites overlap with a tRNA gene (*Figure 5B*). This overlap occurs largely at weak recruitment sites, suggesting that canonical condensin recruitment mechanisms at tRNAs may help to reinforce condensin DC binding on the X chromosome downstream of DCC recruitment at strong sites.

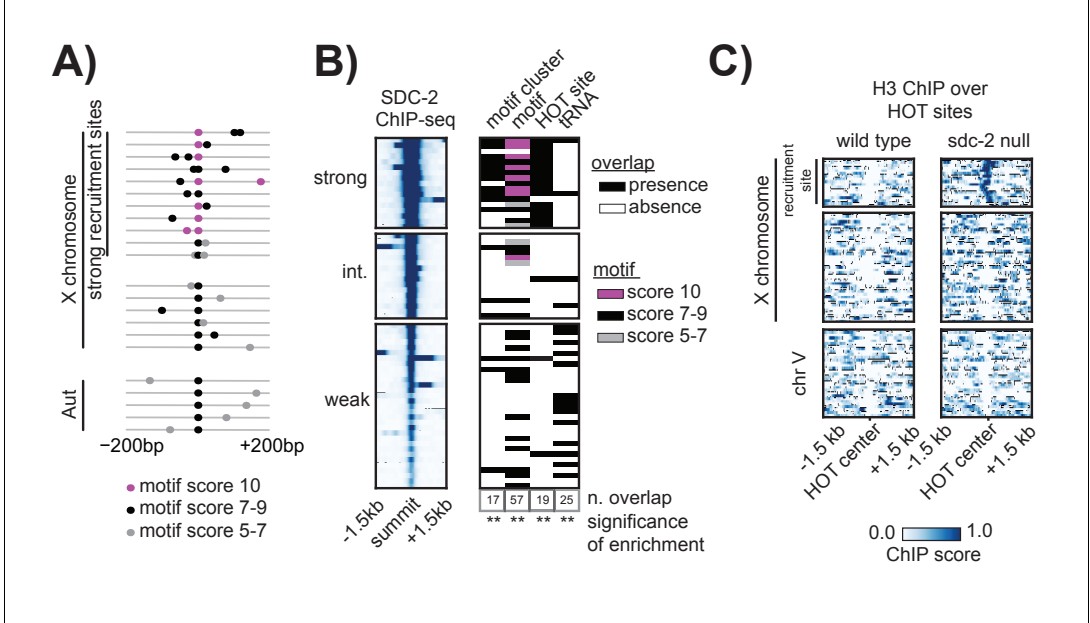

**Figure 5.** Clustered motifs at HOT sites distinguish the X chromosome from the autosomes. (A) A motif cluster is defined as a strong motif (score ≥7) flanked within 200 bp by a second motif. Motif locations plotted for each of the 22 motif clusters: 17 are X- linked (11 at strong recruitment sites, six at weaker recruitment sites) and five are on the autosomes. Plots are centered on the strongest motif in each cluster. Weak motifs (score 5–7) are shown in gray; strong motifs (score 7–9) are shown in black; perfect matches (score = 10) are shown in pink. For simplicity, motif scores (*Supplementary file 1E*) are rounded down to the nearest whole integer. (B) Plot demonstrating overlap between recruitment sites and motif clusters, motif strength, HOT sites, and tRNAs. Black box indicates presence and white box indicates absence of a given overlap. For motifs, pink box indicates presence of a perfect score motif (score = 10); black box indicates presence of a strong motif (score 7–9); grey box indicates presence of a weak motif (score 5–7). Significance of overlap is calculated for the entire set of 64 recruitment sites using permutation test (overlapPermTest function of the regioneR package (*Fruciano et al., 2016*), 100 permutations, ** p-value<0.01). The reported 'n. overlap' refers to the number of indicated genomic annotation that overlaps with at least one recruitment site. Overlap with HOT site is a characteristic of strong recruitment sites. Overlap with tRNAs is a characteristic of weaker recruitment sites. (C) Heatmaps showing H3 ChIP-seq enrichment from wild-type (left) and *sdc-2* null (right) embryos, plotted across a 3 kb window centered on annotated HOT sites. HOT sites on the X are divided into two categories: those that overlap with a recruitment site and those that do not. HOT sites from chromosome V are representative of all autosomal HOT sites. SDC-2 is required for opening chromatin at the HOT DCC recruitment sites on the X.

The following figure supplements are available for figure 5:

**Figure supplement 1.** ChIP-seq data reveals that neither perfect affinity motifs nor motif clustering by itself is sufficient to explain X-specific recruitment of the DCC.

**Figure supplement 2.** Extended analysis of overlap between recruitment sites with each tested annotation.

Interestingly, the set of strong recruitment sites is distinguished by an overlap with HOT sites, defined as genomic regions that are bound by a high number of TFs (38 or more of the 57 TFs assayed in [*Van Nostrand and Kim, 2013*]) without their corresponding motifs (*modENCODE Consortium et al., 2010*) (*Figure 5B*). Among the 17 strong recruitment sites, 16 fall within the boundaries of an annotated HOT site (*Van Nostrand and Kim, 2013*). In *C. elegans*, HOT sites tend to be CpG-rich and typically show characteristics of open chromatin, including nucleo-some depletion (*Chen et al., 2014*). Our H3 data confirms a lack of histone binding at all annotated HOT sites (*Figure 5C*). In the *sdc-2* null mutant, H3 becomes enriched only at those HOT sites that overlap with a recruitment site, indicating that SDC-2 binding is necessary to keep these HOT sites open. Strikingly, while none of the five autosomal motif clusters are HOT, we find that 11 out of 12 motif clusters on the X chromosome both overlap with a HOT site and are characterized as strong recruitment sites. Importantly, this overlap between homotypic motif clusters and HOT sites on the X chromosome fully distinguishes the X from the autosomes.

## Ectopic insertion of *rex-1* fails to robustly recruit the DCC to an autosome

Our analysis has thus far demonstrated that the 17 strong recruitment sites are distinguished from other motif containing sites in the genome by motif clustering, chromatin structure, and overlap with annotated HOT sites. In addition, these strong recruitment sites are bound by SDC-2 in the absence of other DCC subunits, suggesting a hierarchy of DCC recruitment among the recruitment sites. The presence of weaker recruitment sites that are not as easily distinguishable from autosomal sequence suggests that these sites might be active only in the context of the X chromosome. X chromosome dependence of a subset of recruitment sites is consistent with previous work that demonstrated the inability of some large X chromosome duplications to recruit the DCC when detached from the rest of the chromosome (*Blauwkamp and Csankovszki, 2009*).

To test X-dependence of recruitment site activity, we ectopically inserted ~400 bp of recruitment site sequence into autosomal loci that do not bind the DCC in wild-type animals. *Rex-1*, here categorized as an intermediate strength site, was the first recruitment sequence identified by multi-copy extrachromosomal array analysis (*McDonel et al., 2006*). We reasoned that if *rex-1* is sufficient to autonomously recruit the DCC, then inserting 400 bp of *rex-1* sequence into the chromosome II MosSci site should result in ectopic DCC binding. We used ChIP-qPCR to assay for the presence of both SDC-3 (recruiter protein) and DPY-27 (condensin DC) at the autosomal locus.

ChIP-qPCR is a powerful tool for quantitatively determining the amount of antibody-immunoprecipitated DNA. However, the assay is sensitive to the inherent variability of ChIP between biological replicates, necessitating proper controls for normalization. To account for background noise, we used the ΔΔCt method to calculate fold enrichment over input compared to a negative control locus. To account for between-replicate differences in ChIP efficiency, we internally normalized fold enrichment to a positive control (*i.e.* the endogenous locus). Experiments were done in biological triplicate and final ChIP-qPCR results were plotted as percent of endogenous recruitment, allowing comparison across experimental set ups.

ChIP-qPCR results indicate that *rex-1* sequence recruits SDC-3 and DPY-27 at a much lower level when inserted in single copy into chromosome II (*Figure 6A*). Respectively, SDC-3 and DPY-27 qChIP enrichment reaches only 25.79% and 12.40% of endogenous site levels. In contrast, sequence from the strongest recruitment site, *rex-8,* inserted in single copy at the same locus binds SDC-3 and DPY-27 at 46.56% and 21.89% of endogenous *rex-8* levels, respectively. We conclude that *rex-8*, which contains a cluster of three motifs and overlaps with a HOT site in the native X chromosome, is a strong DCC recruiter and is sufficient to ectopically recruit the complex to an autosome, albeit at reduced levels compared to the endogenous site.

Interestingly, when we insert the same 400 bp *rex-1* sequence in single-copy into the X chromosome (at a site that does not normally recruit the DCC) the sequence is able to robustly recruit the DCC, even exceeding endogenous levels (*Figure 6A*). ChIP-seq data also revealed a novel DCC spreading peak just upstream of the *rex-1* insertion site on the X (*Figure 6B*). Although many DCC spreading sites are located in X-linked promoters, this new spreading site is in an un-annotated region. We do not yet understand what may be driving the DCC to spread to this sequence in the ectopic strain where it fails to do so on the wild-type chromosome.

## Multiple recruitment sites cooperate to recruit the DCC to the X chromosome

Single-copy *rex-1* sequence fails to fully recruit the DCC to an autosome, but is sufficient to (both ectopically and endogenously) recruit the DCC to the X chromosome, suggesting X-dependency of *rex-1* activity. DCC recruitment at weaker sites on the X may be dependent on cooperation between multiple recruitment sites. Alternatively, activation of weaker recruitment sites may strictly require the presence of a strong recruitment site. We distinguished between these possibilities by inserting multiple-copies of *rex-1* sequence into two different autosomal loci. If *rex-1* function is dependent on activation by strong recruitment sites, then increasing copy number should have no effect on ectopic recruitment. Alternatively, if multiple recruitment sites function to cooperatively recruit the DCC, then ectopic recruitment should improve upon increased copy-number. Indeed, inserting *rex-1* sequence in three and eight copies respectively increased DPY-27 enrichment to 61.49% and 99.89% of endogenous *rex-1* binding (*Figure 6A*). SDC-3 enrichment similarly increased to 80.98%

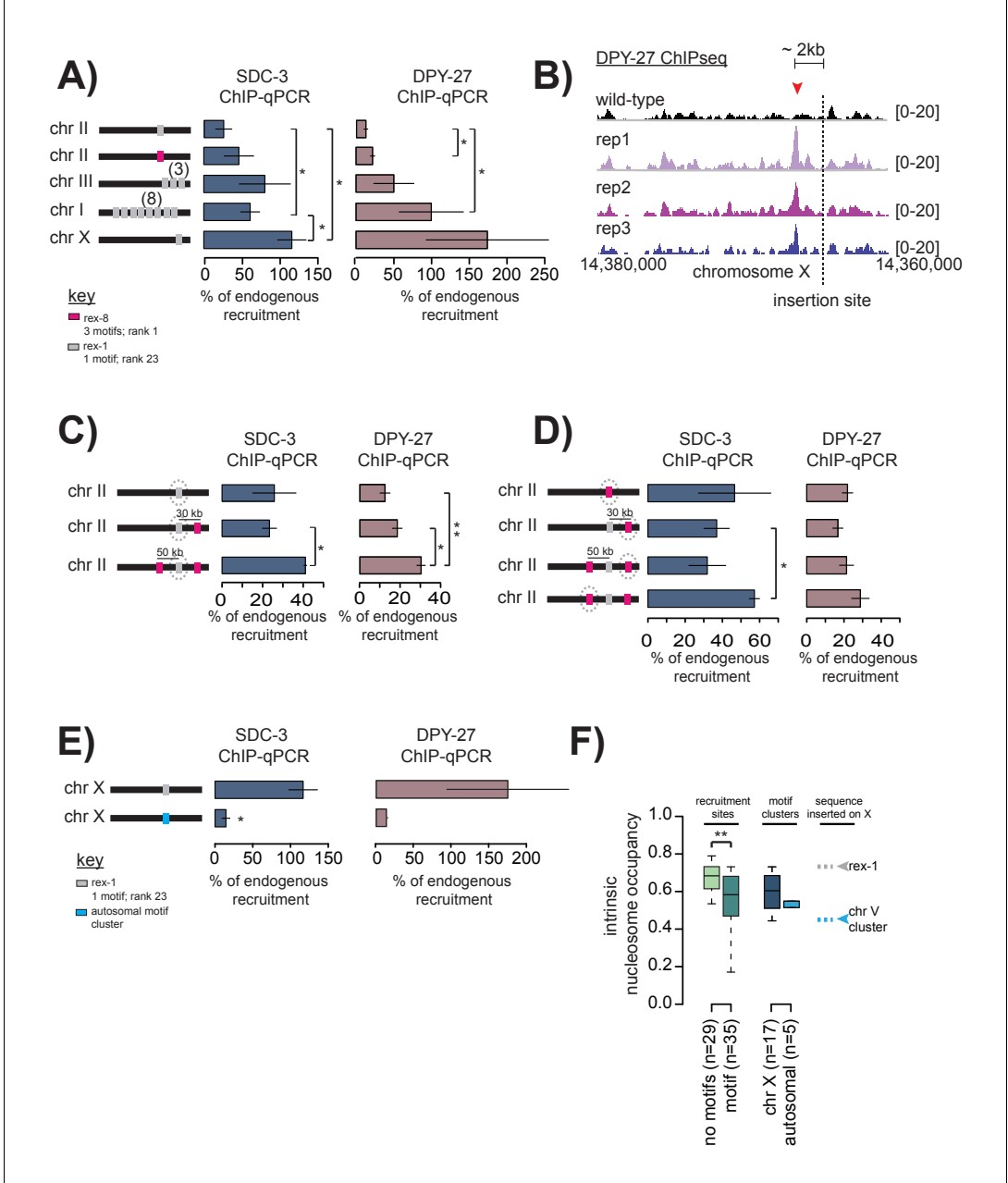

**Figure 6.** Cooperation between recruitment sites is necessary for robust DCC recruitment. (**A**) SDC-3 (left, blue) and DPY-27 (right, brown) ChIP-qPCR data is plotted for the indicated ectopic insertions. ChIP-qPCR is plotted as percent of endogenous recruitment, that is, recruitment at the ectopic site is compared to recruitment at the corresponding endogenous sequence. MosSci was used to insert *rex-1* and *rex-8* sequence into the same locus on chromosome II. Bombardment was used to insert *rex-1* sequence in three-copy on chromosome III and eight-copy on chromosome I. CRISPR was used to insert *rex-1* sequence in single copy on chromosome X. Information on strain generation is available in ***Supplementary file 1A***. Significance is tested using one-tailed t-test assuming unequal variance (*p-value<0.05; **p-value<0.01). (**B**) DPY-27 ChIP-seq data comparing wild-type average and three biological replicates of the ChIP-seq data in the strain bearing an ectopic insertion of *rex-1* in single copy in chromosome X. The insertion site is indicated as the dashed line. Inserted *rex-1* sequence is not shown, as ChIP-seq enrichment at the ectopic sequence is confounded by binding at the endogenous site. Location of the novel spreading peak in the insertion strain is indicated by the red arrowhead. (**C**) As in (**A**), SDC-3 and DPY-27 ChIP-qPCR data is plotted as percent of endogenous recruitment. Dashed circle indicates ectopic site being assayed in ChIP-qPCR. CRISPR was used to insert *rex-8* sequence 30 kb downstream and 50 kb upstream of the ectopic *rex-1* sequence on chromosome II. (**D**) As in (**C**), dashed circle indicates the ectopic site being assayed. (**E**) CRISPR was used to insert a motif cluster from chromosome V into the ectopic chromosome X site. Because the chromosome V site does not normally recruit the DCC, SDC-3 and DPY-27 ChIP-qPCR data is plotted as percent of endogenous *rex-1* recruitment. (**F**) Boxplot indicates intrinsic nucleosome occupancy for a 150 bp window centered on either motif or recruitment site as indicated. Recruitment sites with no motifs (light green, median 0.684) have significantly higher DNA-encoded nucleosome occupancy compared to recruitment sites containing motifs

*Figure 6 continued on next page*

*Figure 6 continued*

(dark green, median 0.583). Motif clusters on the X (dark blue, median 0.604) have higher DNA-encoded nucleosome occupancy compared to motif clusters on the autosomes (light blue, median 0.548). *Rex-1* has a nucleosome occupancy score of 0.732; the chromosome V motif cluster has a nucleosome occupancy score of 0.450).

and 50.08% of endogenous levels (*Figure 6A*). These results indicate that *rex-1* sequence in multiple-copy is sufficient to cooperatively recruit the DCC in the absence of strong recruitment sites. This suggests that the presence of strong recruitment sites in the context of the X chromosome enhances and reinforces cooperativity.

Given that the median distance between recruitment sites on the X is ~90 kb (ranging from 1543 bp to 1.2 Mb), cooperation between sites is likely long-range. To test the ability of recruitment sites to cooperate over long-distances, we used CRISPR to sequentially insert 400 bp of *rex-8* sequence ~30 kb downstream and ~50 kb upstream of the single ectopic *rex-1* that failed to recruit by itself on chromosome II (*Figure 6C*). Notably, we observe similar levels of ectopic recruitment to *rex-8* sequence at three independent insertion loci, suggesting little position effect on recruitment ability (*Figure 6D*). Upon addition of *rex-8* at a single locus ~30 kb downstream, recruitment of DPY-27 to the ectopic *rex-1* increased ~1.5 fold (*Figure 6C*). Insertion of *rex-8* at a second site ~50 kb upstream resulted in further increase in both SDC-3 and DPY-27 binding at the ectopic *rex-1* locus (*Figure 6C*). It is remarkable that, without prior knowledge of the mechanism of long-range cooperativity, insertion of *rex-8* sequence at two distant loci resulted in increased ectopic recruitment to the single copy *rex-1* sequence. These results suggest that strong and weak recruitment sites, dispersed across the length of the X chromosome, cooperatively define a recruiting environment that restricts DCC binding to the X chromosome.

Recent HiC data indicated that interactions between recruitment sites are among the most prominent long-distance contacts on the X chromosome (*Crane et al., 2015*). These interactions are largely DCC dependent as many are lost in an *sdc-2* mutant. We used the same data to determine recruitment site interactions, limiting our analysis to interaction scores above a 99% quantile threshold. Accordingly, we found that 62 out of 64 recruitment sites interact with at least one other recruitment site in the wild-type worm (*Figure 5—figure supplement 2*). This level of site-to-site contact is highly significant (p<0.001, permutation test). In the *sdc-2* mutant, the number of interactions is reduced: 35 out of 64 recruitment sites interact with at least one other site. In *Drosophila*, 3D interaction between high affinity sites facilitates spreading of the dosage compensation complex across the X chromosome (*Ramírez et al., 2015*). One possible explanation for DCC recruitment to weaker sites, especially to those lacking a motif, is that the complex utilizes pre-established spatial proximity to spread from strong sites. Increased DCC binding might reinforce recruitment site interactions, helping to establish the observed X chromosome conformation (*Crane et al., 2015*).

## An autosomal 12 bp motif cluster fails to recruit the DCC to the X chromosome

Because most strong recruitment sites are composed of motif clusters, and because recruitment sites bind the DCC more effectively in the context of the X chromosome, we next tested sufficiency of a motif cluster to recruit the complex when inserted into the X chromosome. To this end, we inserted one of the five identified autosomal motif clusters (containing two motifs) into the X chromosome. We selected this autosomal motif cluster because it contained the strongest scoring motifs (scores 9.5 and 6.25) separated by the shortest genomic distance (68 bp), and was most similar to motif clusters found at strong recruitment sites on the X. To eliminate any potential position effects, we inserted the motif cluster into the same X chromosomal locus where ectopic *rex-1* robustly recruited the DCC (*Figure 6A*). The autosomal motif cluster failed to strongly recruit the DCC to the X chromosome (*Figure 6E*). This result suggests that motif clustering alone is not sufficient for DCC recruitment, even in the context of the X chromosomes. Because we found that strong recruitment sites containing motif clusters overlap with HOT sites and encode for high intrinsic nucleosome occupancy, we reasoned that flanking DNA sequence and/or chromatin structure is important for DCC recruitment. Compared to motif clusters on the X chromosome (median 0.604), autosomal motif clusters have lower intrinsic nucleosome occupancy (median 0.548) (*Figure 6F*). Interestingly,

recruitment sites that do not contain motifs have significantly higher intrinsic nucleosome occupancy than recruitment sites that do contain motifs (*Figure 6F*), suggesting that a specific DNA or chromatin structure is important for recruitment activity.

## Single recruitment site deletions affect DCC binding across large chromosomal domains defined by TAD boundaries

We reasoned that, if recruitment sites act cooperatively to achieve robust and X-specific DCC localization, then deleting a single site on the X chromosome should reveal long-range effects on DCC binding. Using CRISPR, we deleted three recruitment sites on the X chromosome: two strong sites, *rex-40* and *rex-41* are respectively the left and the rightmost strong sites on the X, while the intermediate strength *rex-1* lies ~4 Mb into the X chromosome (*Figure 7—figure supplement 1A*). Deletions were confirmed by Sanger sequencing (*Figure 7—figure supplement 2*). Compared to wild-type, DPY-27 ChIP-seq data in the *rex-41* deletion strain indicates that the DCC binding profile is largely unchanged along the length of the X chromosome (*Figure 7—figure supplement 1B,C*). This suggests that DCC recruitment is a robust process; cooperation between the remaining recruitment sites is capable of countering the effect of a single recruitment site deletion.

Although gross changes are not apparent, we observed a subtle reduction in DCC binding upon *rex-41* deletion. We used a sliding window analysis, an assay more sensitive to slight alterations in DCC enrichment, to determine both the level and significance of change in DCC binding upon recruitment site deletion. Briefly, a 2 Mb window was stepped across each of the ~2500 DPY-27 binding sites on the X chromosome (median step size of 4.8kb). At each step, the mean log2 ChIP-seq ratio between wild-type and recruitment site deletion strain was calculated from all binding sites within the window. To allow comparison across experiments, mean values were normalized using the log2 ratio at the *rex-8* locus and percent deviation from wild-type was plotted. Significance of reduced DCC binding was calculated using students t-test, comparing log2 ratio of DPY-27 binding within a window to binding across the X chromosome. This analysis revealed a significant decrease of ~10–20% in DCC binding across ~1–2 Mb regions surrounding each individual deletion (*Figure 7A*). Additional analyses using 1 Mb and 500 kb windows indicate similar results (*Figure 7—figure supplement 3*). DPY-27 ChIP-seq data from a *set-4* mutant strain (defective in dosage compensation but not in DCC localization) revealed no localized reduction of DCC binding, validating the analysis and confirming that the effect on DCC binding is specific to the deletion strains (*Figure 7—figure supplement 4*). Additionally, mRNA-seq data from the *rex-41* deletion strain indicated a significant increase in local gene expression in the affected region, consistent with reduced DCC binding (*Figure 7B*).

Interestingly, the edges of the regions with significant DCC depletion upon *rex* deletion align with published TAD boundaries on the X chromosome (*Crane et al., 2015*) (*Figure 7A*). This observed proximity, calculated by summing the distance between the edges of the DCC depleted regions and the nearest TAD boundary for all three deletions, is expected by chance less than 7% of the time (permutation test, *Figure 7C*). One possibility is that DCC spreading from recruitment sites is limited to sequences contained within the same TAD. TADs spanning 1–2 Mb have been observed in mammals (*Dixon et al., 2012*; *Nora et al., 2012*) and *C. elegans* (*Crane et al., 2015*; *Gabdank et al., 2016*), while smaller domains have been identified in *Drosophila* (50–100 kb) (*Hou et al., 2012*; *Sexton et al., 2012*). Recent work in *C. elegans* suggests a link between TAD formation and dosage compensation as X chromosome TADs are significantly altered in a dosage compensation mutant (*Crane et al., 2015*). In addition, several recruitment sites coincide with annotated TAD boundaries on the *C. elegans* X, suggesting that strong recruitment at one site may prevent spreading across this site. Alternatively, condensin DC may be loaded along the entire length of the X chromosome before spreading, halting at recruitment sites bound by SDC proteins and resulting in condensin DC enrichment at these sites.

This idea is reminiscent of current models of TAD formation wherein loop extrusion by complexes such as cohesin or condensin form progressively larger loops until stopped by TAD boundaries (*Sanborn et al., 2015*; *Nasmyth, 2001*; *Goloborodko et al., 2016*; *Fudenberg et al., 2016*). In these models, boundary elements bound by CTCF block loop extrusion in an orientation dependent manner (*Fudenberg et al., 2016*; *Rao et al., 2014*). In humans, CTCF sites occur predominantly in a convergent orientation (*Rao et al., 2014*). Though *C. elegans* lack CTCF, we reasoned that motif orientation at recruitment sites might play a similar role. To investigate this possibility, we plotted and

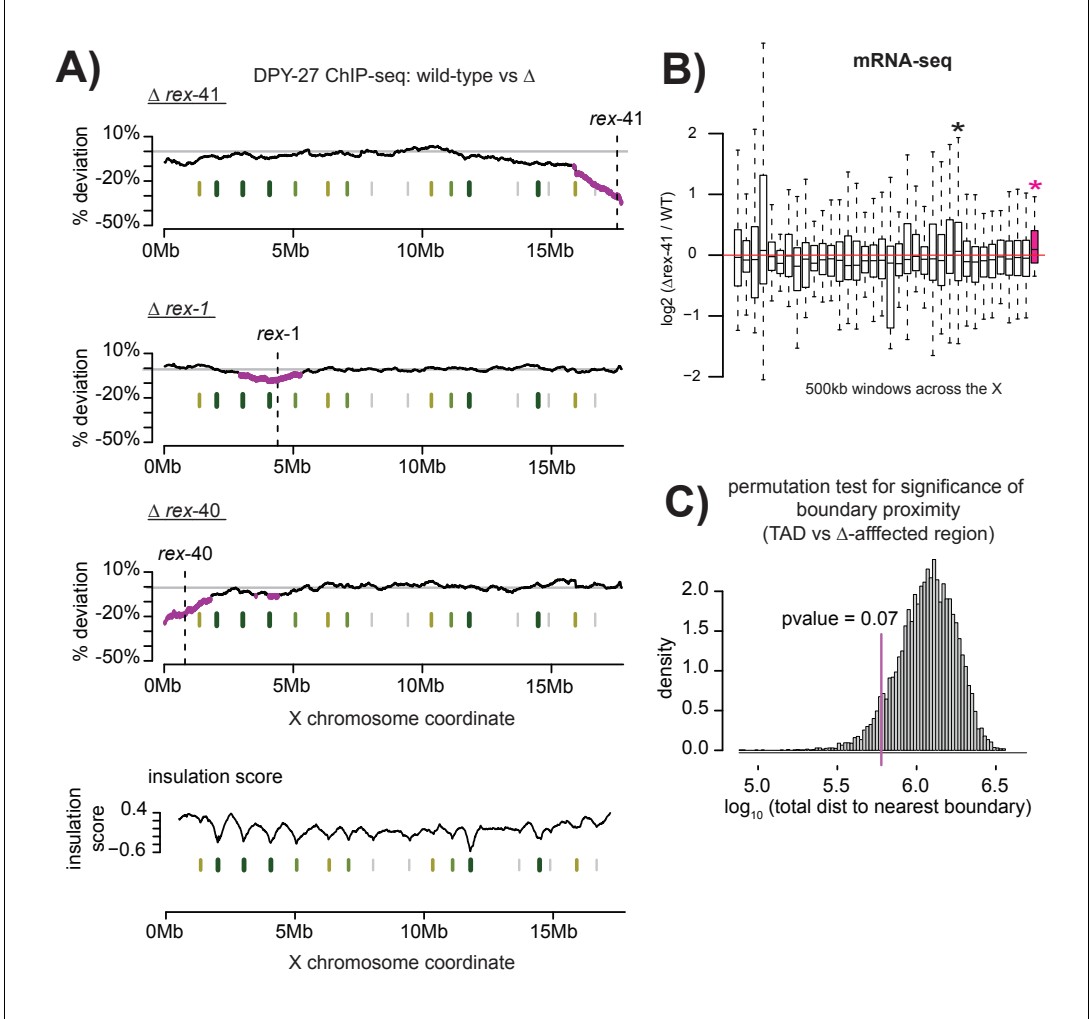

**Figure 7.** Multiple DCC recruitment sites establish DCC binding within defined X chromosomal domains. (**A**) DPY-27 ChIP-seq data was used to calculate percent deviation from wild-type in three different recruitment site deletion strains. Briefly, the log2 ratio between wild-type and deletion strain was calculated for all DPY-27 ChIP-seq peaks and compared to a positive control locus (*rex-8*, the strongest recruitment site). A variable-step sliding window (window size of 2 Mb stepped across each DPY-27 peak) was used to calculate average deviation from wild-type binding. Regions shown in pink have significantly decreased DPY-27 enrichment compared to the rest of the chromosome, p-value<0.05, determined by one tailed students t-test, comparing log2 ratios in an individual window to log2 ratios across the whole X chromosome. In each experiment, the deleted recruitment site is indicated by a dashed line. Green lines (darker with stronger insulation activity) indicate TAD boundaries (as described in [*Crane et al., 2015*]). (**B**) mRNA-seq data comparing *rex-41* deletion strain to wild-type is shown. Boxplot indicates the log2 ratio between deletion and wild-type for 500 kb windows tiled contiguously across the X chromosome. The rightmost window, shown in pink, contains a significant number of genes with increased transcription compared to wild-type (fisher test, p-value=0.0126). Asterisks mark windows with p-value<0.05. (**C**) Summed distance between borders of deletion-affected regions from all three recruitment site deletions in (**A**) and nearest TAD boundary (min: 54 kb, max: 230 kb) is closer than expected by chance alone (average: 313 kb). Shuffling the boundaries of deletion-affected regions (10000 permutations) produced a normal distribution wherein the observed proximity to TAD borders is expected to occur less than 7% of the time by chance. Shown is the distribution of the summed total distance between permuted deletion boundaries and nearest TAD boundary expressed as a $\log_{10}$ value.

The following figure supplements are available for figure 7:

**Figure supplement 1.** Schematic of recruitment site deletions and ChIP-seq data demonstrating DCC recruitment in the deletion strains.

**Figure supplement 2.** Sanger sequencing results from the wild-type and recruitment site deletion strains.

**Figure supplement 3.** Sliding-window analysis of DCC binding change across the X chromosome using different window sizes.

**Figure supplement 4.** Control for sliding-window analysis of DCC binding change across the X chromosome.

*Figure 7 continued*

**Figure supplement 5.** 12-bp motif directionality and recruitment site interactions. .

listed (*Figure 7—figure supplement 5A and B*) the orientation of motifs contained within recruitment sites across the X chromosome, but saw no obvious pattern linking motif orientation to the location of the top HiC contacts. However, it is possible that not all the motifs within a recruitment site are functional. Therefore, given the negative data, we cannot comment on the potential importance of motif orientation in DCC recruitment-mediated TAD organization. Future studies analyzing DCC binding and chromosomal interactions in various recruitment site deletion strains will be needed to elucidate the conservation and diversification of the mechanisms that control condensin recruitment, condensin spreading, and the formation of TAD structures across different systems.

## Discussion

The *C. elegans* DCC is an essential gene regulatory complex, responsible for chromosome-wide repression of X transcription in hermaphrodites (*Plenefisch et al., 1989*; *Kramer et al., 2015*). DCC localization and function is restricted to the hermaphrodite X chromosomes by sex-specific expression of *sdc-2* beginning around the 40 cell stage (*Dawes et al., 1999*). X-specific binding is hypothesized to include two-steps: initial recognition of the X chromosome (dependent on SDC-2, SDC-3, and DPY-30 (*Pferdehirt et al., 2011*; *Dawes et al., 1999*; *Davis and Meyer, 1997*)), followed by DCC spreading (*Ercan et al., 2009*). Although the players were known, the exact mechanism that restricts DCC binding to the X chromosome remained unresolved. Our work provides a clear model for how the DCC specifically binds to the *C. elegans* X chromosomes (*Figure 8*). We propose that initial recruitment involves SDC-2 recognition of nucleosome occupied motif clusters within a HOT context marking a small number of X-specific strong recruitment sites. DCC binding at these strong sites initiates an environment amenable to subsequent DCC recruitment at weaker sites. From the recruitment sites, the DCC is able to spread across large chromosomal domains flanked by TAD boundaries. Hierarchical and cooperative DCC recruitment followed by long-range spreading results in DCC binding along the length of the X chromosome. Notably, hierarchy of DCC recruitment, initiating at the strong sites, requires that only few sites on the X need to be distinguished from autosomes while cooperativity between the recruitment sites ensures robust targeting of the DCC to large X chromosomal domains.

### X-recognition by the DCC occurs at a set of strong recruitment sites marked by motif clustering, intrinsic nucleosome occupancy, and overlap with HOT sites

In the absence of SDC-3, the hermaphrodite specific SDC-2 recognizes and binds only the strongest recruitment sites on the X chromosome (*Figure 2C*). This observation, combined with evidence that DCC enrichment at ectopic *rex-8* (a strong recruitment site) exceeds that of ectopic *rex-1* (an intermediate recruitment site) (*Figure 6A*), is highly suggestive of a recruitment site hierarchy. Recruitment site hierarchy was recently suggested as a mechanism to specify *Drosophila* MSL binding to the male X chromosome. Sites of initial recruitment, termed PionX, contain a novel motif specified by DNA sequence and shape that serve to distinguish the X chromosome from the autosomes (*Villa et al., 2016*). Similarly, we find here that DCC binding on the *C. elegans* X is initiated at a subset of strong recruitment sites distinguished from weaker sites by the presence of 12 bp motif clusters, intrinsic nucleosome occupancy, and overlap with HOT sites. While each of these factors by themselves cannot distinguish the X chromosome from the autosomes, in combination they define a unique set of X-specific DCC recruitment sites. Interestingly, two strong recruitment sites lack motif sequence. It is possible that some yet-unidentified signal (an alternative DNA motif, histone modification, non-coding RNA or long range interaction) may be guiding the DCC to these sites.

Our work indicates that SDC-2 is required for maintaining open chromatin at strong DCC recruitment sites (*Figure 3A*). Additionally, the 12 bp motifs bound by the DCC on the X chromosome are

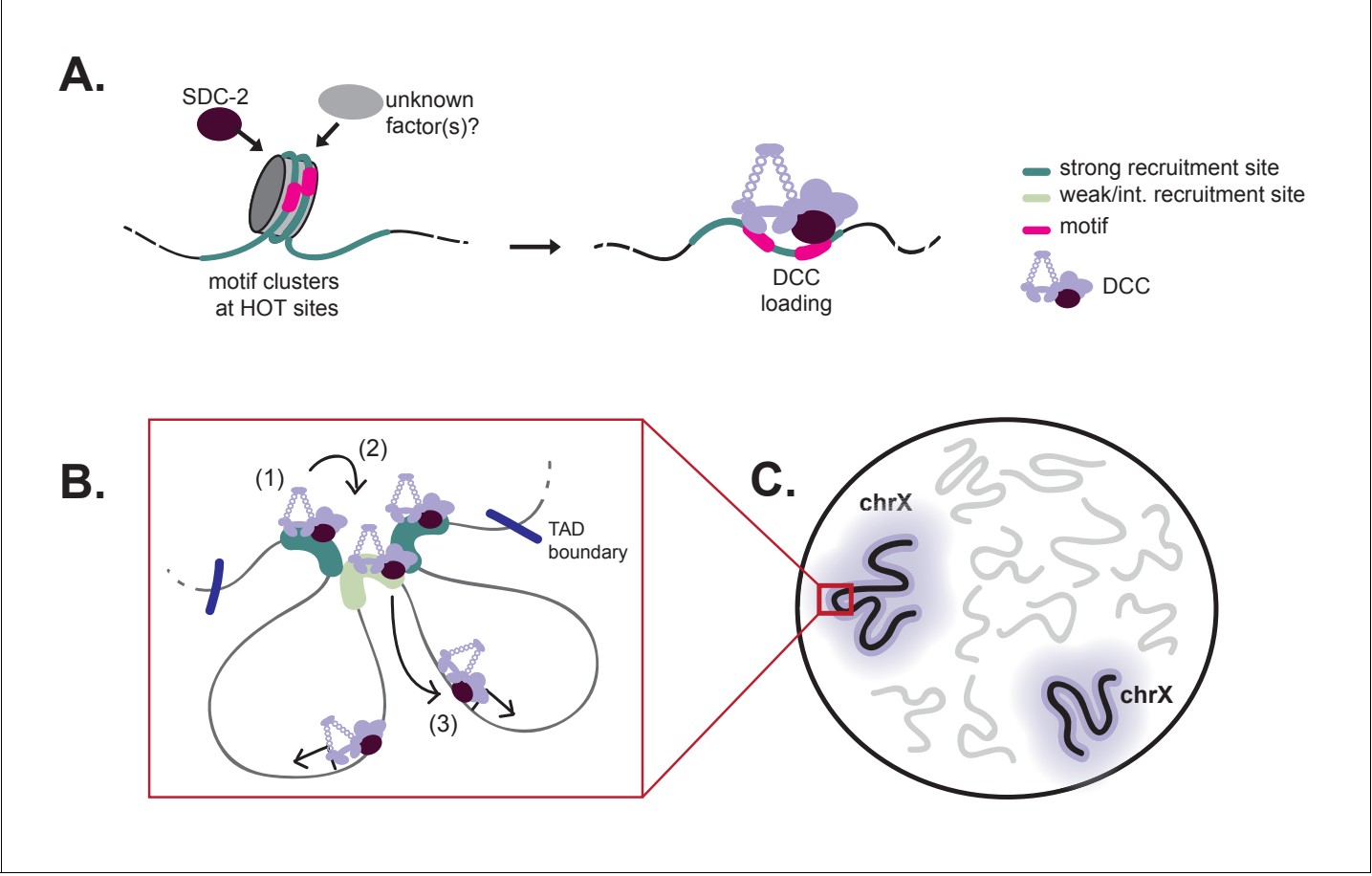

**Figure 8.** A model for X-specific DCC targeting involves SDC-2 recognizing a small number of initial recruitment sites, followed by cooperative recruitment and DCC spreading to distinct chromosomal domains. (A) Initial X-recognition occurs at a subset of strong recruitment sites marked by motif clustering, overlap with HOT sites, and intrinsic nucleosome occupancy encoded in DNA. We hypothesize that nucleosome eviction at these HOT sites may be the function of SDC-2 and may additionally rely on yet-unidentified chromatin remodeling factors. SDC-2 binding is necessary to keep these sites open and allows for DCC binding. It is possible that, prior to *sdc-2* expression, closed chromatin prevents improper DCC loading in early embryogenesis, thus ensuring hermaphrodite specificity. Furthermore, initial loading of DCC at a limited number of X-chromosome specific sites would decrease the evolutionary pressure to select for the 12 bp motif at every recruitment site. (B) Our working model is that DCC recruitment to strong recruitment sites (1) allows activation of the weaker recruitment sites (2) and spreading of the DCC within topologically associating domains (3). Robust recruitment is dependent on long-range cooperativity, which may function through some combination of long-distance physical interactions, high DCC concentration mediated by DCC sequestration, DCC post-translational modifications, and DCC spreading. (C) Cooperation between a set of hierarchical recruitment sites restricts DCC binding to the X chromosomes, establishing large, chromosome-wide transcriptionally repressive domains within the nucleus.

predicted to have higher nucleosome occupancy than motifs that remain unbound (*Figure 4E*). We reason that closed chromatin at the strong recruitment sites prevents the DCC binding prior to *sdc-2* expression, thus ensuring hermaphrodite-specific recruitment of the dosage compensation machinery. This is similar to a recently described mechanism in *Drosophila* wherein the Zelda pioneer protein activates early enhancers by reducing intrinsic nucleosome occupancy and allowing TFs to bind (*Sun et al., 2015*). It is not yet clear if SDC-2 acts as a pioneer factor, mediating DCC recruitment by increasing DNA accessibility. Alternatively, some yet unidentified factor(s) might work with or in advance of SDC-2 to recognize and evict nucleosomes. For instance, overlap of clustered 12 bp motifs with annotated HOT sites may reflect interaction between SDC-2 and various TFs in establishing open chromatin at recruitment sites. Recent work in fission yeast has proposed a role for transcriptional co-activators Gcn5 and RSC in evicting nucleosomes prior to condensin binding (*Toselli--*

*Mollereau et al., 2016*). Notably, it remains unclear how SDC-2 is able to recognize the 12 bp motif, as it contains no known DNA binding domains.

## Multiple recruitment sites cooperate to robustly recruit the DCC to large X chromosomal domains

Ectopic insertion of recruitment site sequence (*Figure 6A,C*) and deletion of recruitment sites on the X (*Figure 7A*) revealed an important feature of DCC recruitment: the cooperativity of recruitment sites over long distance. In single copy, both *rex-1 and rex-8* recruited the DCC to an autosomal locus at a fraction of their activity on the X. While it is possible that reduced ectopic recruitment is a consequence of chromosomal location, our data shows little evidence for positional effects. We found that *rex-1* sequence inserted in three copies at a secondary autosomal locus similarly showed reduced *rex-1* activity compared to the endogenous site (*Figure 6A*), and ectopic recruitment to *rex-8* sequence was largely consistent between three chromosome II insertion sites (*Figure 6D*). Additionally, others have recently reported that *rex-32* (a strong recruitment site) recruits the DCC to multiple sites across the genome (*Wheeler et al., 2016*). In light of these observations, lack of ectopic DCC recruitment to single-copy autosomal *rex-1* sequence suggests that recruitment activity of this intermediate strength recruitment site is X-dependent, requiring the cooperation of nearby sites. The importance of recruitment site cooperativity is supported by previous observations that several X chromosome duplications containing recruitment sites do not recruit the DCC when detached from the X (*Csankovszki et al., 2004*; *Blauwkamp and Csankovszki, 2009*). Here, we found that adding multiple recruitment site sequence either nearby (*Figure 6A*) or at a distance (*Figure 6C*) increased ectopic *rex-1* recruitment, while recruitment site deletions resulted in reduced DCC binding across large chromosomal domains (*Figure 7A*). Together these results demonstrate the functional importance of cooperation between multiple recruitment sites in establishing robust DCC binding to the X chromosome.

## A distinction between condensin DC and SDC subunits of the DCC

Notably, our ChIP-qPCR data suggests a distinction between recruitment of the condensin DC core and the SDC subunits. In single-copy ectopic *rex-1* and *rex-8* insertions, SDC-3 was recruited better than DPY-27 (*Figure 6A*). Adding additional recruitment sites nearby resulted in comparatively larger gains in DPY-27 enrichment, suggesting a greater role for cooperativity in the recruitment of condensin DC. Previous work has also demonstrated a difference in the X-chromosome binding pattern between condensin DC and the SDC proteins (*Ercan and Lieb, 2009*). By comparing DCC subunit binding between a well-defined site of spreading and a site of recruitment, it was shown that condensin DC spreads more effectively than both SDC-2 and SDC-3 (*Ercan and Lieb, 2009*). Lastly, our ChIP-seq data indicates the presence of the DCC recruitment module (SDC-2, SDC-3, and DPY-30) at scattered sites across all autosomes that fail to recruit condensin DC. These differences, combined with a lack of biochemical evidence for an intact DCC, are highly suggestive of distinct mechanisms governing recruitment of the SDC proteins and of condensin DC. While SDC proteins recognize and target specific sites on the X and the autosomes, condensin DC binding is much more dependent on cooperativity of multiple recruitment sites on the X.

## Role of long-range cooperativity and nuclear localization in X-specific DCC targeting

Because recruitment sites are separated by a median distance of ~90 kb (544 bp – 1.2 Mb) in the native X chromosome, we expect cooperation to act over long-distances. But what is the mechanism governing recruitment site cooperativity? We can imagine several possibilities, none of which are mutually exclusive. First, cooperative 3D interactions between recruitment sites (*Crane et al., 2015*) might reinforce DCC binding by increasing the local concentration of condensin DC. Previous work indicates that DCC binding alters the nuclear positioning of the X chromosome (*Sharma et al., 2014*) and compacts it by ~40% (*Lau et al., 2014*). It has been postulated that condensin complexes induce chromosome compaction by introducing and stabilizing long-range contacts between distant chromosomal loci (*Cuylen and Haering, 2011*; *Thadani et al., 2012*). Indeed, condensin, similar to cohesin, can entrap DNA, mediating interactions between distant chromosomal sites (*Cuylen et al., 2011*, *Cuylen et al., 2013*). Recent work in fission yeast demonstrated the ability of condensin to

drive long-range associations across distances spanning 100 kb to 1 Mb (*Kim et al., 2016*). Similarly, SDC-2 depletion, which results in lack of condensin DC recruitment to the X, reduced the number of 3D contacts between recruitment sites (*Crane et al., 2015*). It is possible that DCC binding strengthens pre-existing X chromosome contacts while simultaneously inducing conformational changes that mediate interactions between recruitment sites and that these direct 3D interactions allow for the cooperative recruitment of the DCC.

We envision that physical interaction between the DCC recruitment sites (*Crane et al., 2015*) paired with X chromosome compaction (*Lau et al., 2014*) may generate a nuclear compartment of high DCC concentration. Sequestration of the DCC to a compacted X chromosome territory may function in restricting re-loading of the DCC to the X chromosome. Potential recruitment sites on the autosomes would not benefit from this increased concentration of the DCC and would fail to maintain DCC binding. This model is reminiscent of transcription factories wherein stable nuclear compartments containing a high local concentration of Pol II lower the threshold for preinitiation complex formation for all nearby promoters (*Cook, 2010*). A surplus of DCC would reinforce complex binding at strong sites while simultaneously allowing for DCC recruitment to weaker recruitment sites. Clustering of recruitment sites in the nucleus to generate a compartment for DCC loading is an effective and elegant mechanism by which to cooperatively and specifically recruit the DCC to the X chromosomes.

Second, it was shown that in vivo depletion of SUMO significantly disrupts DCC recruitment to the X chromosomes (*Pferdehirt and Meyer, 2013*). Although it is not clear if this effect is direct, western blot and mass-spectrometry analyses indicated that SDC-3, DPY-27, and DPY-28 are targets of SUMOylation (*Pferdehirt and Meyer, 2013*). DPY-28, the non-SMC subunit that functions in both condensin I and condensin DC, is SUMOylated only in the context of the DCC (*Pferdehirt and Meyer, 2013*). It may be that successive SUMOylation of DCC subunits at multiple recruitment sites along the X somehow modulates DCC dynamics on the chromatin, perhaps increasing DCC affinity at recruitment sites, thereby reinforcing X specificity.

Finally, recruitment site cooperativity may be mediated through condensin DC spreading. It has been proposed that after recruitment, the DCC spreads along the length of the chromosome (*Ercan et al., 2009*). Spreading may be achieved by sliding of condensin DC ring along the chromatin. Alternatively, condensin DC may hop to distant chromosomal loci via long-range interactions between recruitment and spreading sites. The mechanism of DCC spreading remains unknown. Notably, the size of the region affected by a recruitment site deletion is similar to the distance that the DCC can spread from the X chromosome into autosomal sequence (*Ercan et al., 2009*). In three X;A fusion chromosomes, levels of DCC enrichment on autosomal sequence was shown to be inversely proportional to the distance from the fused end of the X chromosome. Similarly, reduced DCC binding near a deleted recruitment site is also dependent on proximity to the deletion (*Figure 7A*). The observed pattern of decreased DCC binding in the recruitment site deletion strains suggests that DCC spreading is restricted to TAD boundaries. In mouse embryonic cells, deletion of a boundary element within the *Hox* locus resulted in spreading of active chromatin across the border (*Narendra et al., 2015*). Spreading of SMC complexes such as cohesin and condensin, as well as other chromatin regulating proteins may be limited by the TAD boundaries. The DCC is also implicated in the creation of the TAD boundaries themselves (*Crane et al., 2015*), suggesting that the mechanism by which the DCC is loaded onto and spreads across the chromosome may influence boundary formation. Mechanistically, spreading may serve to both establish and reinforce DCC binding sites across the length of the X chromosome.

It is not clear if cooperativity also contributes to the X-specific recruitment of the *Drosophila* MSL complex. Although the strategies are distinct, X-specific binding of the *C. elegans* DCC and the *Drosophila* MSL show significant parallels. In both, a DNA sequence motif is important for recruitment, but is not specific to the X chromosome (*Conrad and Akhtar, 2012*; *Alekseyenko et al., 2012*, *Alekseyenko et al., 2008*). Recent work has indicated that GA repeat expansion around an 8 bp motif core helps specify MSL binding at high affinity sites, mediated through CLAMP zinc-finger binding (*Kuzu et al., 2016*). This dinucleotide expansion may serve to distinguish X and autosomal motif sites (*Kuzu et al., 2016*). However, just like the DCC recruiters (SDC-2, SDC-3 and DPY-30), CLAMP binds to autosomal loci independent of MSL (*Soruco et al., 2013*). Previous studies found that ectopic insertion of *Drosophila* high affinity MSL recruitment sites are sufficient to recruit the MSL complex to an autosome (*Kelley et al., 1999*; *Alekseyenko et al., 2008*). Although the level of

this ectopic recruitment was not precisely quantified, it results in spreading to nearby active genes where the complex functions to up-regulate transcription (*Ramírez et al., 2015*). It would be interesting to determine if the specific recruitment of *Drosophila* MSL to the X chromosome also involves cooperativity between high affinity sites on the X.

### Specificity of condensin targeting to chromosomes

Condensin complexes are conserved from yeast to humans, functioning in both chromosome condensation during cell division and gene regulation during interphase (*Hirano, 2016*; *Cobbe et al., 2006*; *Hirano, 2012*; *Lau and Csankovszki, 2014*). Studies in several organisms have revealed high-resolution condensin binding patterns, demonstrating that condensin complexes bind to distinct intergenic sites that are enriched for promoters and enhancers (*Kranz et al., 2013*; *Pferdehirt et al., 2011*; *Kim et al., 2013*; *Dowen et al., 2013*). In yeast, the TATA box binding protein (TBP), the histone acetyltransferase, Gcn5, and the RSC remodeling complex have been shown to recruit condensin to its chromosomal binding sites (*Toselli--Mollereau et al., 2016*; *Iwasaki et al., 2015*). While the metazoan condensin recruiters remain largely uncharacterized (*Piazza et al., 2013*), our previous work indicated that, similar to condensin DC, the *C. elegans* condensin II complex binds to DCC recruitment sites in an SDC-2 dependent manner (*Kranz et al., 2013*). It remains unclear if, similar to condensin DC, recruitment is cooperative for canonical condensins. Interestingly, in yeast, tRNA genes that recruit condensin cluster in three-dimensional space and this clustering is, in part, mediated by condensin itself (*Haeusler et al., 2008*). Furthermore, yeast ribosomal DNA (rDNA) repeat region robustly recruits condensin in a FOB1 dependent manner (*Johzuka et al., 2006*). It is possible that the repetitive structure of the rDNA locus allows for cooperative recruitment through multiple tandem copies of the recruiting sequence. Notably, the weaker DCC recruitment sites are enriched at tRNA gene loci, suggesting that once the initial X-specific sites recruit the DCC, common mechanisms of condensin loading help to maintain DCC binding on the X chromosomes.

### Long-range cooperation in genome organization and transcriptional regulation

Our model postulates that long-range cooperation between a hierarchical set of recruitment sites allows for the robust and specific targeting of the DCC to the X chromosomes without the necessity of evolving and maintaining domain-specific motifs at every recruitment site. This model has important implications for TF targeting and domain-level transcriptional regulation in eukaryotic genomes. Recent work in various organisms has begun to focus on the functional role of cooperation between regulatory elements. For example, in *Drosophila*, tandem gene duplications often exhibit higher than 2-fold increased transcription, potentially due to the additive effects of transcription factors binding identical sites on both gene copies (*Loehlin and Carroll, 2016*). In human B cells, long-range enhancer interactions help explain lineage-specific control of Tcrb recombination (*Proudhon et al., 2016*). In yeast, cooperation between mating-type silencers leads to transcriptional repression (*Boscheron et al., 1996*). Finally, similar to how weak recruitment sites bind the DCC only in multiple copies or in the context of the X chromosome, it has been proposed that some *Drosophila* Polycomb domains with weak binding sites may recruit the Polycomb group protein, PHO, through cooperation between high affinity sites (*Schuettengruber et al., 2014*). From yeast, to flies, to humans, cooperative interactions in 3D space are increasingly being used to explain genome-targeting specificities of proteins important in processes ranging from recombination to transcriptional silencing. We add to this list a clear paradigm for studying the genetic mechanisms behind long-range cooperativity: recruitment of the dosage compensation complex to the *C. elegans* X chromosomes.

## Materials and methods

### Worm strains and growth

Wild-type (N2), ERC06 (knuSi254[SNP400bprex-1, unc-119(+)] II; unc-119(ed3) III), ERC08 (knuIs6 [pSE-02(400bprex-1SNP), unc-119(+))] I; unc-119(ed3) III), ERC09 (knuIs7[pSE-02(400bprex-1SNP), unc-119(+))] III; unc-119(ed3) III), ERC38 (ers30[delX:17544437–17544484, delX:17545624–

17545624]), ERC51 (ersIs17[SNP400bp_rex1, X:14373144]), ERC54 (ers20[delX:4394846–4396180]; knuIs6[pSE-02 (400bprex-1SNP), unc-119(+))] I; unc-119(ed3) III), ERC64 (ers28[delX:806628–806813]; knuIs6[pSE-02(400bprex-1SNP), unc-119(+))] I; unc-119(ed3) III), ERC61 (ersSi25 [wt400bp_rex8, cb-unc-119(+)] II; unc-119(ed3) III), ERC62 (ersIs26[X:11093923-11094322[rex-8], II: 8449965); knuSi254[SNP400bprex-1, unc-119(+)] II; unc-119(ed3) III), ERC63 (ersIs27[X:11093923-11094322[rex-8], II:8371600, II:8449968); knuSi254[SNP400bprex-1, unc-119(+)] II; unc-119(ed3) III), ERC66 (ersIs30[V:14040305-14040785[chrV_clusteredmotifs], X:14373144]), TY2205 (her-1(e1520) sdc-3(y126) V; xol-1(y9) X), TY1072 (her-1(e1520) V; sdc-2 (y74) X), MT14911 (set-4 (n4600) II).

Information regarding strain names, genotype, and generation of all new strains, used in this study can be found in *Supplementary file 1A*. Briefly, for all insertion strains, 400 bp of either *rex*-1 (chrX:4395400–4395799, WS220) or *rex*-8 sequence (chrX:11093923–11094322, WS220), (centered at the DCC ChIP binding peak summit and containing recruitment motif sequence) was integrated into an ectopic genomic location using MosSci (*Zeiser et al., 2011*), bombardment, or CRISPR (*Dickinson and Goldstein, 2016*) based techniques. In the bombardment strains (ERC08 and ERC09), insertion sites were mapped using genomic DNA paired end Illumina sequencing data. We isolated discordant read pairs that connected bombarded plasmid sequence to a specific region of the genome, eliminating spurious connections by comparing against paired-end sequencing data obtained from wild-type N2 genomic DNA. Insertion copy-numbers were calculated using qPCR analysis of genomic DNA, normalizing against a negative control region in the genome, and using the single-copy insertion strain (ERC06) generated by MosSCI as a positive control reference. Deletion strains were generated using CRISPR; sgRNAs were designed flanking recruitment site sequence, ensuring deletion of all identified 12 bp motifs contained within. PCR followed by Sanger sequencing was used to validate insertions and deletions. Deleted sequences and the motifs contained within can be found in *Figure 7—figure supplement 3*. Strains were maintained at 20°C on NGM agar plates using standard *C. elegans* growth methods. Mixed stage embryos were isolated from gravid adults by bleaching. For ChIP experiments, embryo samples were fixed by treating with 2% formaldehyde for 30 min. For RNA experiments, embryo samples were resuspended in 10 volumes of Trizol.

## Antibodies, ChIP-seq, and ChIP-qPCR

*Supplementary file 1B* includes the list of the ChIP experiments and information on the antibodies. For ChIP, embryos were washed and dounce homogenized in FA buffer (50 mM HEPES/KOH pH 7.5, 1 mM EDTA, 1% Triton X-100, 0.1% sodium deoxycholate; 150 mM NaCl). Sarkosyl (0.1% sodium lauroyl sarcosinate) was added before sonicating to obtain chromatin fragments between 200 and 800 bp in length. As in *Ercan et al., 2007*, 1 to 2 mg of embryo extract and 3 to 8 ug of antibody was used per ChIP. For library preparation, half of the ChIP DNA was ligated to Illumina TruSeq adapters and amplified by PCR. Library DNA between 250 and 500 bp was gel purified. Single-end 50 bp sequencing was performed using the Illumina HiSeq-2000 at the New York University Center for Genomics and Systems Biology, New York, NY. ChIP-qPCR was performed using 4–5% of each ChIP sample and the corresponding diluted input DNA, as described in *Mukhopadhyay et al. (2008)*. KAPA SYBR FAST 2X qPCR Master Mix (Kapa Biosystems, MA) was used in 20 μl reactions and performed on the Roche LightCycler 480 system. Data (input and chip sample) were used to calculate $\Delta\Delta Ct$ between experimental and control loci. Briefly, $\Delta\Delta Ct = $ (experimental Ct –control Ct) $_{ChIP}$ –(experimental Ct –control Ct) $_{input}$. Enrichment over background for both endogenous and ectopic loci was calculated as $2^{-\Delta\Delta Ct}$. Percent endogenous recruitment was then calculated by taking the ratio of ectopic enrichment over endogenous enrichment and multiplying by 100. Because background enrichments are variable between biological replicates, $\Delta\Delta Ct$ and percent endogenous were calculated individually for each replicate. DNA sequence for the qPCR primers and product sizes are given in *Supplementary file 1C*.

## ChIP-seq data processing

We aligned 50 bp single-end reads to *C. elegans* genome version WS220 using bowtie version 1.0.1 (*Langmead et al., 2009*), allowing two mismatches in the seed, returning only the best alignment, and restricting multiple alignments to, at most, four sites in the genome. Mapped reads from ChIP and input were used to call peaks and obtain read coverage per base using MACS version 1.4.2

(*Zhang et al., 2008*) with default parameters. Coverage per base was normalized to the genome-wide median coverage (excluding the mitochondrial chromosome). Final ChIP enrichment scores per base were obtained by subtracting matching input coverage. We merged replicates by taking the average ChIP enrichment scores at every position in the genome. For the *sdc-2* and *sdc-3* null mutants (each of which have altered X:A ratios), X and autosomal reads were processed separately and final ChIP enrichment scores were combined after normalization. Raw data files and wiggle tracks of ChIP enrichment per base pair are provided at NCBI's Gene Expression Omnibus (*Edgar et al., 2002*) database under accession number [GEO: GSE87741]. Basic statistics and the GEO number for each ChIP-seq data are provided in *Supplementary file 1B*.

To determine a set of binding sites for each protein, we imposed a majority rule using the biological replicates, as explained in *Kranz et al., 2013*. Briefly, reads from the replicates were combined using samtools version 0.1.19 (*Li et al., 2009*) and MACS was used to call peaks at a stringent p-value cutoff e-10. Only those peaks from the combined set that were also present in the majority of the individual replicates were included in the final peak set. Peak summits were determined as the position with the maximum ChIP enrichment score in the combined average. To avoid penalization of long peaks with multiple summits, peaks were split into smaller peaks using PeakAnalyzer version 1.4 (*Salmon-Divon et al., 2010*), with the minimum height being equal to the median coverage at all determined summits of the given data set and a separation float of 0.85. The analysis scripts are written in R and Perl and are available as *Source code 1*.

## mRNA-seq

Worms were collected and stored in ten volumes of Trizol (Invitrogen). Samples were freeze-cracked five times and RNA purification was done according to the manufacturer protocol. Isolated RNA was cleaned up using the Qiagen RNeasy kit. The mRNA was purified using Sera-Mag Oligo (dT) beads (Thermo Scientific) from at least 1 ug of total RNA. Stranded mRNA-seq libraries were prepared based on incorporation of dUTPs during cDNA synthesis using a previously described protocol (*Parkhomchuk et al., 2009*). Single-end 50 bp sequencing was performed using the Illumina HiSeq-2000. Reads were aligned to genome version WS220 with Tophat version 2.0.0 (*Trapnell et al., 2012*) using default parameters. For each biological replicate, read numbers and mapping percentages (which refer to the percentage of unique reads with at least one alignment) can be found in *Supplementary file 1B*. Count data was calculated using HTSeq version 0.6.1 (*Anders et al., 2015*) and normalized using the R package DESeq (*Anders and Huber, 2010*). The raw reads and counts can be obtained from GEO under accession number [GEO: GSE87741].

## Identification of recruitment sites

To estimate the strongest binding sites (foci [*Ercan et al., 2007*]) for SDC-2, a mixture distribution with two components was assumed: one for the majority of binding sites with low coverage and one for binding sites with an extremely high coverage at the peak summits. To differentiate between binding sites with strong and low coverage, the coverage at the peak summits of all binding sites was used to estimate the parameters of a mixture distribution with the function normalmixEM from the mixtools package version 1.0.3 (*Benaglia et al., 2009*) in R version 3.2.1 (*R Core Team, 2015*). SDC-2 peaks overlapping with SDC-3, DPY-30, and DPY-27 but not overlapping with H3K4me3 peaks were successively determined using the command intersectBed of the BEDTools suite version 2.12.0 (*Quinlan and Hall, 2010*). Deeptools version 2.2.3 (*Ramírez et al., 2016*) was used to rank order recruitment sites based on SDC-2, SDC-3, DPY-30, and DPY-27 ChIP enrichment for a 3 kb window centered on the peak summit. Recruitment sites were clustered into three categories (strong, intermediate, and weak) using k-means clustering (n = 3). Recruitment site coordinates and rankings can be found in *Supplementary file 1D*.

## Motif analysis

DNA sequence of ±100 bp around the summit of the top 200 SDC-2 ChIP-seq binding peaks was used to identify potential binding motifs using MDScan (*Liu et al., 2002*). The position weight matrix of the top motif was used to scan and score the whole genome using TRAP (*Thomas-Chollier et al., 2011*). Resulting TRAP affinity scores were standardized to a 0 to 10 scale. Motif locations and scores can be found in *Supplementary file 1E*.

## Acknowledgements

We thank NYU-CGSB High Throughput Sequencing Facility for sequencing and raw data processing. We thank Knudra Trangenics, Cecilia Pellegrini and Sofija Miljovska for their contribution to making strains. Some strains were provided by the CGC, which is funded by NIH Office of Research Infrastructure Programs (P40 OD010440). Research reported in this publication was supported by NIGMS of the National Institutes of Health under award number R01GM107293.

## Additional information

### Funding

| Funder | Grant reference number | Author |
|---|---|---|
| National Institute of General Medical Sciences | R01GM107293 | Sarah Elizabeth Albritton Anna-Lena Kranz Lara Heermans Winterkorn Lena Annika Street Sevinc Ercan |

The funders had no role in study design, data collection and interpretation, or the decision to submit the work for publication.

### Author contributions

SEA, Conceptualization, Data curation, Formal analysis, Validation, Visualization, Methodology, Writing—original draft, Writing—review and editing; A-LK, Data curation, Formal analysis; LHW, LAS, performed experiments; SE, Conceptualization, Formal analysis, Supervision, Funding acquisition, Investigation, Writing—original draft, Writing—review and editing

### Author ORCIDs

Sarah Elizabeth Albritton, http://orcid.org/0000-0001-6093-9560
Sevinc Ercan, http://orcid.org/0000-0001-7297-1648

## Additional files

### Supplementary files

• Source code 1. Perl and R scripts for data analysis. Perl scripts making up the ChIP-seq analysis pipeline are provided. R script and command lines for sliding window analysis of DCC binding changes across the X chromosome are provided.

• Supplementary file 1. (A) Strain information. The names and the genotypes of each strain are given in the sheet 'strains'. Sequences for the sgRNAs and the primers used to generate the homology-mediated repair oligos are listed in the sheet 'primers.' Deletion and insertion coordinates as well as the inserted sequence are also indicated. (B) Data summary. The GEO accession numbers, total and mapped read counts, and antibody information are provided for the ChIP, RNA, and DNA-seq data. (C) qChIP primers. Primer sequences used for the qChIP experiments are provided. (D) Recruitment site information. The coordinates of the 64 DCC recruitment sites, the motifs contained within, and the recruitment strength category of each site are provided. (E) Motifs across the genome. Motif sequence and coordinates for all 12 bp motifs identified across the genome are provided. Supplementary file 1F: Annotation information. The annotation sources that are used in this manuscript.

### Major datasets

The following dataset was generated:

| Author(s) | Year | Dataset title | Dataset URL | Database, license, and accessibility information |
|---|---|---|---|---|
| Albritton SE, Kranz A, Winterkorn LH, | 2016 | Cooperation between a hierarchical set of recruitment sites | https://www.ncbi.nlm. nih.gov/geo/query/acc. | Publicly available at NCBI Gene Expression |

| Street LA, Ercan S | specifically targets the C. elegans dosage compensation complex to the X chromosomes. | cgi?acc=GSE87741 | Omnibus (accession no: GSE87741) |

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
