## [Decision Letter]

Thank you for submitting your article "Cooperation between a hierarchical set of recruitment sites targets the X chromosome for dosage compensation" for consideration by *eLife*. Your article has been reviewed by three peer reviewers, one of whom is a member of our Board of Reviewing Editors, and the evaluation has been overseen by Kevin Struhl as the Senior Editor. The reviewers have opted to remain anonymous.

The reviewers have discussed the reviews with one another and the Reviewing Editor has drafted this decision to help you prepare a revised submission.

Summary:

In this manuscript Albritton et al. investigate the targeting mechanisms for the dosage compensation machinery in *C. elegans*. The nature of the sequences/chromatin features that are required for the targeting of the DCC have been sought, and although rex sequences that are enriched on the X chromosome have indeed been found, unique X-linked sequences do not appear to exist and the question of DCC targeting specificity is still very much open. The authors use ChIP-seq data in embryos, for various DCC components (including SDC-2, SDC-3, DPY-30, DPY-27) and H3K4me3 to identify potential features that can explain X-specificity. The present study analyzed ChIP-seq data for SDC-2, SDC-3, DPY-30, DPY-27 and H3K4me3 in *C. elegans* embryos to identify the 64 strongest binding sites (for DCC subunits that do not overlap with H3K4me3), all of which are on the X, and then analyzed the features of the 17 strongest binding sites in that set of 64. Their analysis supports the existence of a relatively small number of recruitment sites that share several features which together confer X specificity. Those features include a high-scoring binding sequence, other nearby binding sequences, a HOT domain, and intrinsic DNA-driven nucleosome occupancy. The authors also perform elegant functional studies – deletions and knock-ins of predicted recruitment sites. The emerging model is that SDC-2 and other DCC subunits are initially recruited to relatively few strong recruitment sites, spread to nearby weaker recruitment sites via cooperativity, and can further spread within TADs. The paper is well written and represents a significant advance with novel mechanistic insights into the process of dosage compensation in *C. elegans*, although some points need to be addressed:

Main Points:

1) Figure 2 shows reduced association of SDC-2 with X recruitment sites in *sdc-3* mutants. The authors interpret this as persistent association with the strong sites and loss of association with the intermediate and weak sites and use this as evidence for a hierarchy of recruitment between strong and weak sites. Another interpretation is that SDC-2 association is equivalently reduced at all sites. A possible way to examine this further is via a scatter plot and correlation analysis of SDC-2 binding in wild type versus SDC-2 binding in the *sdc-3* mutant. That might help illuminate if in the *sdc-3* mutant the level of SDC-2 binding is reduced equivalently or not on strong, intermediate, and weak recruitment sites. It could also be used to address if there are new SDC-2 binding sites in *sdc-3* compared to wild type.

2) In Figure 5, how were HOT sites defined i.e. how many TFs bound?

3) The authors should be more cautious regarding the claim that SDC-2 opens chromatin for initial DCC recruitment. They base themselves on the presence of H3 in *sdc-2* null mutant at the rex sites (Figure 3). However, in the absence of SDC-2, this site is not bound by the DCC, which allows it to be chromatinized (hence H3 presence). With the current data, it is difficult to distinguish between chromatin opening and simple occupancy of the site by the DCC (in wild-type animals).

4) Related to the above point, can the authors distinguish if Sdc-2 acts to load the DCC as opposed to trigger focal enrichment (ChIP-seq peaks) at the Sdc-2 binding sites? This could be addressed by analyzing the total number of condensin ChIP-seq tags across the X chromosome (not just peaks) in Sdc-2 (and -3) mutants. If numbers are similar this implies that condensin can still be loaded on the X even without Sdc-2, and that Sdc-2 is therefore likely to be important for concentrating condensin at its binding sites, rather than loading it. Alternatively, if condensin is still present diffusely on the X in *Sdc-2* mutants, then the authors might wish to consider a model whereby long range cooperativity arises by pairs of Sdc-2 bound sites blocking the translocation of condensin, leading to focal condensin enrichment, accumulation of chromatin loops and formation of a TAD boundary (see below).

5) The part of the paper dealing with TADs raises some very interesting and important issues but merits some clarification and further analyses:

– In Figure 7, a horizontal line indicating zero would be helpful. For the *rex-41* deletion, the DPY-27 signal appears to be consistently below zero for maybe 5MB, across several strong TAD boundaries. Similarly for *rex-40*.

– The authors chose a 1MB window to average the DPY-27 signal, and report a "significant decrease in DCC binding across ~1-2 MB regions surrounding each deletion". But with a large 1 MB window the smallest regions that can be detected as DCC depleted seem to be 1-2MB. If DCC depletion were highly localized, for example lost only on one strong DCC site, it is conceivable that it would significantly lower the DPY-27 signal on smoothing windows 1MB in both directions, even though DPY-27 reduction is narrowly localized. It would be helpful to see the data analyzed with significantly smaller smoothing windows in addition to the large-window analysis.

– The authors state that to be able to compare the DPY-27 signal between wild type and each rex deletion strain, they first normalized each to their respective rex-8 locus DPY-27 signal. Are the authors confident that DPY-27 signal at the rex-8 locus is unaffected in the rex site deletions, e.g. that there is not a global reduction of DPY-27?

– In Figure 7. A horizontal line indicating zero would again be helpful. In addition to the right-most box, 10th from the right box also has an asterisk and is therefore significantly upregulated? The p-values seem not to be very strong. Were they corrected for multiple hypothesis testing? The authors tested >30 boxes, which would make a corrected p-value significantly higher.

– In Figure 7. This panel addresses an important point. Please explain what this analysis shows.

6) An analysis of rex site orientation would be very informative. Current models of TAD formation suggested by the Mirny and the Liebermann-Aiden laboratories (PMIDs 27210764, 27224481, 26499245), based on an early model from Kim Nasmyth suggest a loop extrusion model involving complexes (e.g. SMC) that are stopped by proteins bound at oriented boundaries (CTCF sites in mammalian cells). The parallel for rex sites with the mammalian Cohesin / CTCF system is striking. The authors could explore the orientation of the motifs presented in Figure 4 at recruitment sites (cf. Rao et al. Cell 2014) to see whether pairs of neighboring strong / weak sites tend to be in a specific configuration (convergent, divergent or tandem). In particular they could assess whether orientation of sites inside clusters correlates with DCC recruitment strength and could potentially explain some results presented in Figure 6 (the orientation is not described in these experiments). This could also be investigated for the TAD boundaries that are found to interact. In this regard, the Crane paper suggests that the strongest rex sites are associated with TAD boundaries, while the model in this paper suggests that strong sites are located inside TADs. A line-up of the relative positions of TAD boundaries and strong/weak recruitment sites will shed light on this. And Figure 8 should be accordingly modified. It would also be informative to show condensin enrichment over oriented strong / weak sites.

7) More detailed information on the bioinformatics approaches – normalization procedures – for the different data sets should be provided. The use of the mixtools package should be better described. Ideally the authors could provide scripts as part of the paper, or even the entire analysis as in some recent cases (PMID 27919068).

8) The authors should correct/modify several statements:i) In Abstract: "SDC-2 is required to open chromatin" – please moderate this statement.ii) The second paragraph of the Introduction – the authors introduce dosage compensation and focus only on male / female differences – they should also discuss X:A ratio dosage compensation.iii) In the same paragraph the authors state that in each case of dosage compensation a protein complex is specifically targeted to the X in only the sex where is regulates transcription: this is not true in mammals, Xist RNA is expressed from the X only when there are 2 or more X chromosomes, regardless of whether there is a Y present or not: thus in XO females, as in XY males, there is no targeting of Xist and its protein partners to the X.iv) The authors state in this paragraph that the targeting of the DCC involves a 2-step strategy – recruitment and spreading – again this is a proposal – it should not be stated as a fact.v) At the end of their introduction the authors mention TADs – they do not describe what these correspond to at all – and it is not clear that TADs are universal entities (for example, see review by Dekker and Heard 2015), particularly in *C. elegans* (Crane et al., 2015) where they only appear to exist on the dosage compensated X. The authors should explain and cite appropriate references.

[Editors' note: further revisions were requested prior to acceptance, as described below.]

Thank you for resubmitting your work entitled "Cooperation between a hierarchical set of recruitment sites targets the X chromosome for dosage compensation" for further consideration at *eLife*. Your revised article has been favorably evaluated by Kevin Struhl (Senior editor), and three reviewers, one of whom is a member of our Board of Reviewing Editors.

The manuscript has been improved but there are a few remaining issues that need to be addressed before acceptance, as outlined in the reviewers comments.

One particular point concerns the new H3 data provided in Figure 3—figure supplement 1. This data is not strong and the H3 plots should probably be left out. Also, in Figure 7, it is important to explain if the vertical pink line represents 1 or all 3 deletion-affected regions. In Figure 7—figure supplement 5, the TAD boundaries as defined by Crane et al. should also be shown on the same graph. Once these changes and the other minor points outlined below are taken into account your paper will be ready for acceptance.

Reviewer #1:

The authors have addressed most of the issues raised and have improved their manuscript according to many of the suggestions from the reviewers.

There are just a couple of points that are discussed in the response to reviewers points but that merit inclusion in the revised text:

Point 4 "While we cannot exclude the possibility that, prior to *sdc-2* expression, some DPY-27 is loosely associated with all chromosomes, the current data suggest that SDC-2 is required for X-specific loading of condensin DC."

Reviewer #2:

This revised submission is certainly improved and almost ready for publication.

The new analysis shown in Figure 2—figure supplement 1 is nice. We are a bit surprised that in panel B all of the intermediate sites and some of the strong sites display more loss of SDC-2 in the *sdc-3* mutant than the average across the X. Is the X average of -1.280 in panel B driven by many regions of the X having low levels of SDC-2 in both N2 and *sdc-3* (the leftward cloud of gray spots in panel A)? In the main text, I suggest changing "To eliminate the possibility" to "To test the possibility" to avoid having a bias toward certain expected results.

Are the authors convinced by Figure 3—figure supplement 1 that H3 is enriched at the strong sites in EE? The EE profiles look noisy and would need error bars. Most of the EE samples generated by modENCODE were not that early (not necessarily skewed for <40-cell stage as mentioned in subsection “SDC-2 is required to maintain open chromatin at strong recruitment sites, which display intrinsic DNA-encoded nucleosome occupancy*”*, and later-stage embryos containing more nuclei might dominate ChIP patterns), so using them as representative of pre-DC seems risky. Finally, why are the scales different in Figure 3 and Figure 3—figure supplement 1?

In the legend to Figure 7, please explain the pink vertical line. Does it represent 1 or all 3 deletion-affected regions?

In the legend to Figure 7—figure supplement 5, please explain that the numbers in parentheses in panel A are the ranks. In the legend, the sentence "Recent HiC data" needs to be fixed.

Reviewer #3:

This revised version of the manuscript by Albritton et al. integrates all major modifications which we requested on the first review round. Technically, the paper is currently very good and the interpretations of the data careful.

I was however a little frustrated by the analysis of the TAD boundary/rex site orientation and the subsequent discussion, which is limited to the old loading-spreading model.

---

## [Author Response]

*Main Points:*

*1) Figure 2 shows reduced association of SDC-2 with X recruitment sites in sdc-3 mutants. The authors interpret this as persistent association with the strong sites and loss of association with the intermediate and weak sites and use this as evidence for a hierarchy of recruitment between strong and weak sites. Another interpretation is that SDC-2 association is equivalently reduced at all sites. A possible way to examine this further is via a scatter plot and correlation analysis of SDC-2 binding in wild type versus SDC-2 binding in the sdc-3 mutant. That might help illuminate if in the sdc-3 mutant the level of SDC-2 binding is reduced equivalently or not on strong, intermediate, and weak recruitment sites. It could also be used to address if there are new SDC-2 binding sites in sdc-3 compared to wild type.*

We followed this excellent suggestion and added Figure 2—figure supplement 1. In part A, plotting the average SDC-2 ChIP enrichment within a 200 bp window sliding 100 bp along the genome, in N2 vs TY2205 (*sdc-3* mutant), we observe a striking difference between strong recruitment sites and weaker recruitment sites. Across the X chromosome, the median SDC-2 enrichment ratio between the *sdc-3* mutant and N2 is 0.420. This relationship suggests a consistent decrease in SDC-2 binding across the length of the X chromosome, in agreement with what is known in the literature. We use this median value across the X as a background difference in SDC-2 binding between N2 and our mutant. Focusing next on windows overlapping with weak recruitment sites, a simple linear regression model gives a slope of 0.16. Thus, compared to the background, weak recruitment sites exhibit markedly reduced levels of SDC-2 binding in the *sdc-3* mutant. In contrast, a linear regression model comparing SDC-2 enrichment at the 200bp windows overlapping strong recruitment sites gives a slope of 28.190. This suggests a marked increase in SDC-2 binding at strong recruitment sites in the *sdc-3* mutant compared to background. Figure 2—figure supplement 1 part B includes a scatter plot comparing SDC-2 enrichment in N2 to the log2 change in enrichment in the mutant. Here we focus on just the 64 recruitment sites. Each data point represents a 40bp window around the recruitment site summit. The median log2 ratio across the X chromosome is -1.280. We see that weaker recruitment sites exhibit a markedly lower log2 ratio (median -3.042). Strong recruitment sites have a median log2 ratio of -1.057. In summary, in plotting the SDC-2 enrichment across the X chromosome, we find further support for our interpretation: in an *sdc-3* mutant, SDC-2 is able to identify and bind strong recruitment sites. Lastly, we see no evidence for new SDC-2 binding sites on the X chromosome.

*2) In Figure 5, how were HOT sites defined i.e. how many TFs bound?*

For these analyses, we utilized a previously defined list of annotated HOT sites (see Van Nostrand et al., 2013). Briefly, they identified 296 HOT sites as those bound by 38 or more of the 57 assayed transcription factors (> 65%). We edited the main text to explicitly state this definition, referencing the original paper. References for all genomic annotations can be found in Supplemental File 1F 6.

*3) The authors should be more cautious regarding the claim that SDC-2 opens chromatin for initial DCC recruitment. They base themselves on the presence of H3 in sdc-2 null mutant at the rex sites (Figure 3). However, in the absence of SDC-2, this site is not bound by the DCC, which allows it to be chromatinized (hence H3 presence). With the current data, it is difficult to distinguish between chromatin opening and simple occupancy of the site by the DCC (in wild-type animals).*

To address this question, we downloaded early embryo H3 ChIP-seq data from modENCODE (GEO Accession: GSE50259). We use the early embryo data as a proxy to measure H3 occupancy before the onset of DCC binding and sdc-2 expression, with the caveat that the early embryo collections are skewed for less than 40-cells, but not absolutely, so some of the cells may already have some SDC-2 binding. Comparing this to mixed stage embryos (which skew toward more than 100-cell stages with full SDC-2 expression) showed a quantitative difference. H3 enrichment across a 1000bp window, centered on the recruitment site summit show that H3 enrichment is higher in the early embryos compared to the mixed stage embryos. In mixed stage embryos, H3 enrichment is reduced in a narrow window around the summit. This observation supports our hypothesis that strong recruitment sites are occupied before SDC-2 binding. This analysis has been added as Figure 3—figure supplement 1. We commented on this analysis in the main text.

*4) Related to the above point, can the authors distinguish if Sdc-2 acts to load the DCC as opposed to trigger focal enrichment (ChIP-seq peaks) at the Sdc-2 binding sites? This could be addressed by analyzing the total number of condensin ChIP-seq tags across the X chromosome (not just peaks) in Sdc-2 (and -3) mutants. If numbers are similar this implies that condensin can still be loaded on the X even without Sdc-2, and that Sdc-2 is therefore likely to be important for concentrating condensin at its binding sites, rather than loading it. Alternatively, if condensin is still present diffusely on the X in Sdc-2 mutants, then the authors might wish to consider a model whereby long range cooperativity arises by pairs of Sdc-2 bound sites blocking the translocation of condensin, leading to focal condensin enrichment, accumulation of chromatin loops and formation of a TAD boundary (see below).*

The suggested analysis (assessment of condensin DC loading in the mutant background) would require ChIP-seq of the condensin DC-specific subunit, DPY-27, in wild-type as well as the two *sdc* mutant animals. We do not currently have that data. However, the literature indicate that prior to the onset of sdc-2 expression, DPY-27 is diffuse in the nucleus and onset of *sdc-2* expression triggers DPY-27 association with the X chromosome (PMID 10364546). While we cannot exclude the possibility that, prior to sdc-2 expression, *some* DPY-27 is loosely associated with all chromosomes, the current data suggest that SDC-2 is required for X-specific loading of condensin DC.

5) The part of the paper dealing with TADs raises some very interesting and important issues but merits some clarification and further analyses:

*– In Figure 7, a horizontal line indicating zero would be helpful. For the rex-41 deletion, the DPY-27 signal appears to be consistently below zero for maybe 5MB, across several strong TAD boundaries. Similarly for rex-40.*

These are added.

*– The authors chose a 1MB window to average the DPY-27 signal, and report a "significant decrease in DCC binding across ~1-2 MB regions surrounding each deletion". But with a large 1 MB window the smallest regions that can be detected as DCC depleted seem to be 1-2MB. If DCC depletion were highly localized, for example lost only on one strong DCC site, it is conceivable that it would significantly lower the DPY-27 signal on smoothing windows 1MB in both directions, even though DPY-27 reduction is narrowly localized. It would be helpful to see the data analyzed with significantly smaller smoothing windows in addition to the large-window analysis.*

We have added analyses with 500 kb, 1Mb and 2Mb windows in Figure 7—figure supplement 3. Using different size windows show roughly the same pattern of decreased enrichment, and the significant windows are roughly the same size. Furthermore, we are using a variable step sliding window analysis wherein a fixed size window is stepped one peak at a time across the ~2500 DPY-27 peaks on the X. The median distance between the peaks is 4.8kb (with a range of 114bp to 60kb), which is the resolution of the statistics test. Because we are only looking at changes in DPY-27 enrichment at the DPY-27 peaks across the X, window sizes must be large enough to ensure they contain a sufficient number of peaks to allow calculation of p-value using student’s t-test. At a window size of 1Mb, there are between 45 and 183 DPY-27 peaks per window. At a window size of 500kb, there are between 10 and 100 DPY-27 peaks per window. If we were to further reduce the window size to 100kb, there would be only between 2 and 32 DPY-27 summits per window, making a significance test very difficult.

*– The authors state that to be able to compare the DPY-27 signal between wild type and each rex deletion strain, they first normalized each to their respective rex-8 locus DPY-27 signal. Are the authors confident that DPY-27 signal at the rex-8 locus is unaffected in the rex site deletions, e.g. that there is not a global reduction of DPY-27?*

It is difficult to ascertain absolute level of DPY-27 binding from ChIP data, but the worms with the recruitment site deletions are rigorous and wild type looking, suggesting there is no global changes in dosage compensation. Since ChIP efficiency differs between replicates, to compare data between multiple recruitment site deletion strains, we chose to normalize data to the *rex*-8 locus, which is one of the strongest recruitment elements that is far from the deleted loci. Normalizing the data in this manner affects neither the overall trend of the plot nor the calculated p-values. Even if it were the case that the *rex-*8 had slightly reduced levels of DPY-27, we would be underreporting the percent deviation values without any affect on the localized depletion. For these reasons, we feel confident in presenting the data as the percent deviation normalized to the *rex-*8 locus.

*– In Figure 7. A horizontal line indicating zero would again be helpful. In addition to the right-most box, 10th from the right box also has an asterisk and is therefore significantly upregulated? The p-values seem not to be very strong. Were they corrected for multiple hypothesis testing? The authors tested >30 boxes, which would make a corrected p-value significantly higher.*

The DCC has only a 2-fold repressive effect on transcription, and the percent reduction in binding is also modest, so we expect the changes to be very subtle. Here, multiple test correction is not appropriate because we are actually doing one test: is the last window on the X containing the deleted site show significant reduction in expression? This is not the same type of question inherent to multiple testing problems in which the tester is interested in *finding the windows* that show significant reduction in expression.

*– In Figure 7. This panel addresses an important point. Please explain what this analysis shows.*

The analysis shows that the borders of the deletion-affected region and the TAD boundaries are close to each other. Since the effect of each recruitment site deletion is stronger closest to the deleted locus, and tempers off near the TAD boundaries, we interpret this result as TAD boundaries stop the spreading of condensins.

*6) An analysis of rex site orientation would be very informative. Current models of TAD formation suggested by the Mirny and the Liebermann-Aiden laboratories (PMIDs 27210764, 27224481, 26499245), based on an early model from Kim Nasmyth suggest a loop extrusion model involving complexes (e.g. SMC) that are stopped by proteins bound at oriented boundaries (CTCF sites in mammalian cells). The parallel for rex sites with the mammalian Cohesin / CTCF system is striking. The authors could explore the orientation of the motifs presented in Figure 4 at recruitment sites (cf. Rao et al. 2014) to see whether pairs of neighboring strong / weak sites tend to be in a specific configuration (convergent, divergent or tandem). In particular they could assess whether orientation of sites inside clusters correlates with DCC recruitment strength and could potentially explain some results presented in Figure 6 (the orientation is not described in these experiments). This could also be investigated for the TAD boundaries that are found to interact. In this regard, the Crane paper suggests that the strongest rex sites are associated with TAD boundaries, while the model in this paper suggests that strong sites are located inside TADs. A line-up of the relative positions of TAD boundaries and strong/weak recruitment sites will shed light on this. And Figure 8 should be accordingly modified. It would also be informative to show condensin enrichment over oriented strong / weak sites.*

We have added relevant analyses as Figure 7—figure supplement 5. We investigated if the motif orientation is important for the formation of recruitment site-mediated TAD boundaries across the *C. elegans* X chromosome. We plotted the motif orientation as it relates to DCC binding, both within individual recruitment sites and between neighboring recruitment sites. We could not find an obvious interaction pattern with respect to motif orientation. However, this does not mean that motif orientation does not matter, as we do not know which of the motifs are leading the binding and the interactions.

*7) More detailed information on the bioinformatics approaches – normalization procedures – for the different data sets should be provided. The use of the mixtools package should be better described. Ideally the authors could provide scripts as part of the paper, or even the entire analysis as in some recent cases (PMID 27919068).*

The Materials and methods section details our normalization procedures and the data analysis steps, and we added the module used in mixtools package, which is referenced if readers require for further information. Our scripts are mostly hardcoded for our HPC system, and not streamlined into a pipeline, but we have a collection of the scripts and a detailed protocol for their usage, and combined them into a Supplementary file. A note to this end is added to the Materials and methods section.

*8) The authors should correct/modify several statements:i) In Abstract: "SDC-2 is required to open chromatin" – please moderate this statement.*

Abstract edited.

*ii) The second paragraph of the Introduction – the authors introduce dosage compensation and focus only on male / female differences – they should also discuss X:A ratio dosage compensation.*

This information is added.

*iii) In the same paragraph the authors state that in each case of dosage compensation a protein complex is specifically targeted to the X in only the sex where is regulates transcription: this is not true in mammals, Xist RNA is expressed from the X only when there are 2 or more X chromosomes, regardless of whether there is a Y present or not: thus in XO females, as in XY males, there is no targeting of Xist and its protein partners to the X.*

We are focusing on the wild type conditions, but of course, X chromosome dosage compensation can be uncoupled from sex under mutant conditions in mammals, *D. melanogaster* and in *C. elegans*. For instance *her-1* mutant XO animals do not recruit the DCC to their X chromosome, despite being phenotypically female. And *tra-1* XX males are able to dosage compensate. We find it is useful to describe the elegance by which the targeting of dosage compensation machinery to the X chromosome happens in one sex in the three described systems.

*iv) The authors state in this paragraph that the targeting of the DCC involves a 2-step strategy – recruitment and spreading – again this is a proposal – it should not be stated as a fact.*

This is corrected.

*v) At the end of their introduction the authors mention TADs – they do not describe what these correspond to at all – and it is not clear that TADs are universal entities (for example, see review by Dekker and Heard 2015), particularly in C. elegans (Crane et al., 2015) where they only appear to exist on the dosage compensated X. The authors should explain and cite appropriate references.*

Text and references have been added.

[Editors' note: further revisions were requested prior to acceptance, as described below.]

*The manuscript has been improved but there are a few remaining issues that need to be addressed before acceptance, as outlined in the reviewers comments.*

*One particular point concerns the new H3 data provided in Figure 3—figure supplement 1. This data is not strong and the H3 plots should probably be left out.*

We left these plots out based on the recommendation of the reviewers. Accordingly, Figure 3—figure supplement 1 is removed.

*Also, in Figure 7, it is important to explain if the vertical pink line represents 1 or all 3 deletion-affected regions.*

The vertical pink line represents all 3 deletion-affected regions. This information is added to the text and the figure legend.

*In Figure 7—figure supplement 5, the TAD boundaries as defined by Crane et al. should also be shown on the same graph.*

The TAD boundaries are added to 7S5 A.

*Reviewer #1:*

There are just a couple of points that are discussed in the response to reviewers points but that merit inclusion in the revised text:

*Point 4 "While we cannot exclude the possibility that, prior to sdc-2 expression, some DPY-27 is loosely associated with all chromosomes, the current data suggest that SDC-2 is required for X-specific loading of condensin DC."*

This sentence is added to the text.

*Reviewer #2:*

*The new analysis shown in Figure 2—figure supplement 1 is nice. We are a bit surprised that in panel B all of the intermediate sites and some of the strong sites display more loss of SDC-2 in the sdc-3 mutant than the average across the X. Is the X average of -1.280 in panel B driven by many regions of the X having low levels of SDC-2 in both N2 and sdc-3 (the leftward cloud of gray spots in panel A)?*

Yes, we think that this is driven by background signal.

*In the main text, I suggest changing "To eliminate the possibility" to "To test the possibility" to avoid having a bias toward certain expected results.*

We made the suggested change to the text.

*Are the authors convinced by Figure 3—figure supplement 1 that H3 is enriched at the strong sites in EE? The EE profiles look noisy and would need error bars. Most of the EE samples generated by modENCODE were not that early (not necessarily skewed for <40-cell stage as mentioned in subsection “SDC-2 is required to maintain open chromatin at strong recruitment sites, which display intrinsic DNA-encoded nucleosome occupancy”, and later-stage embryos containing more nuclei might dominate ChIP patterns), so using them as representative of pre-DC seems risky. Finally, why are the scales different in Figure 3 and Figure 3—figure supplement 1?*

We have removed the analyses based on the comment that these embryos are not that early, and the summary recommendation from the reviewers (see above).

*In the legend to Figure 7, please explain the pink vertical line. Does it represent 1 or all 3 deletion-affected regions?*

This is clarified based on the summary recommendation from the reviewers (see above).

*In the legend to Figure 7—figure supplement 5, please explain that the numbers in parentheses in panel A are the ranks. In the legend, the sentence "Recent HiC data" needs to be fixed.*

This is fixed.

*Reviewer #3:*

This revised version of the manuscript by Albritton et al. integrates all major modifications which we requested on the first review round. Technically, the paper is currently very good and the interpretations of the data careful.

*I was however a little frustrated by the analysis of the TAD boundary/rex site orientation and the subsequent discussion, which is limited to the old loading-spreading model.*

We have added a sentence about the possibility that condensin DC is loaded everywhere and somehow accumulates at the *rex* sites. “Alternatively, condensin DC may be loaded along the entire length of the X chromosome before spreading, halting at recruitment sites bound by SDC proteins and resulting in condensin DC enrichment at these sites.”